

# Monitoring soil moisture from middle to high elevation in Switzerland: Set-up and first results from the SOMOMOUNT network

Cécile Pellet[1], Christian Hauck[1]

5  [1]Department of Geosciences, University of Fribourg, Fribourg, 1700, Switzerland

*Correspondence to*: Cécile Pellet (cecile.pellet@unifr.ch)

**Abstract.** Besides its important role in the energy and water balance at the soil-atmosphere interface, soil moisture can be a particular important factor in mountain environments since it influences the amount of freezing and thawing in the subsurface and can affect the stability of slopes. In permafrost areas, it is strongly linked to the ground ice content and by 10  this modifies the characteristics and behaviour of periglacial landforms.

In spite of its importance, the technical challenges and its strong spatial variability usually prevents soil moisture from being measured operationally at high and/or middle altitudes. This study describes the new Swiss soil moisture monitoring network SOMOMOUNT launched in 2013 consisting in six entirely automated soil moisture stations distributed along an altitudinal gradient between the Jura Mountains and the Swiss Alps, ranging from 1205m to 3410m elevation. In addition to the 15  standard instrumentation comprising Frequency Domain Reflectometry (FDR) and Time Domain Reflectometry (TDR) sensors along vertical profiles, soil probes and meteorological data are available at each station.

In this contribution we will present a detailed description of the SOMOMOUNT instrumentation and calibration procedures. Additionally, the data collected during the three first years of the project will be discussed in relation to their altitudinal distribution. Clear differences in soil moisture patterns are visible between sites with permanently and seasonally frozen as 20  well as unfrozen ground conditions and can be related to several factors such as the subsurface composition (organic versus mineral), the elevation and the snow cover characteristics.

**Keywords**: Soil moisture, monitoring network, TDR, FDR, mountain, elevation gradient, seasonal frost, permafrost

## 1 Introduction

25  Soil moisture is a key factor controlling the energy and water exchange processes at the soil-atmosphere interface as well as the physical properties of the subsurface such as heat capacity and thermal conductivity (for a review see e.g. Seneviratne et al., 2010). In 2010 soil moisture was classified as an Essential Climate Variable (ECV) by the Global Climate Observing System (GCOS) and has thus to be continuously and globally monitored. Even though the number of soil moisture networks is globally increasing, it is still far from being standardised, coordinated or spatially representative. Coordination efforts are





however increasing with the International Soil Moisture Network being the largest soil moisture data source to date (Dorigo et al., 2011).

Existing soil moisture monitoring network have many different foci, such as the validation of remote sensing products (e.g. Bircher et al., 2012; Rautiainen et al., 2012), the investigation of hydrological processes at hillslope scale (e.g. Brocca et al., 2007; Martini et al., 2015) or catchment scale (e.g. Bogena et al., 2010) as well as the study of land-atmosphere interactions (e.g. Hauck et al., 2011; Krauss et al., 2010; Mittelbach et al., 2011). In the interest of representativeness for large scale studies and easy implementation most of the current monitoring networks are located at middle to low elevation. In Switzerland, the long term monitoring network SwissSMEX was initiated in 2008 and is composed of 16 stations distributed across the Swiss plateau and other low elevation regions (Mittelbach and Seneviratne, 2012).

In mountain environments, where the ground is affected by seasonal and permanently frozen conditions, soil moisture and water flow can be particularly crucial. Its effect on the stability of slopes and the thermal and kinematic characteristics of periglacial landforms was highlighted in several observation and modelling studies (e.g. Boike et al., 2008; Hasler et al. 2011; Hinkel et al., 2001; Krautblatter et al. 2012; Scherler et al., 2010; Streletskiy et al. 2014; Westermann et al., 2009; Zhou et al. 2015). A general overview of the interactions between hydrological, mechanical and ecological processes in frozen grounds is given by Hayashi (2013).

However, soil moisture measurements in mountainous areas are technically challenging because of the often coarse blocky substrate, the temperatures below the freezing point and the remoteness of the sites, which also adds difficulties regarding energy supply and data transfer. They are therefore also more costly to implement. Furthermore, among the numerous well established in situ soil moisture monitoring devices (e.g. Hillel, 2004; Robinson et al., 2008; Vereecken et al., 2014), only few have been tested in such conditions (e.g. Pellet et al., 2016; Rist and Phillips, 2005, Zhou et al., 2015). Thus measurements in mountainous terrains are currently restricted to uncoordinated and project based installations (e.g. Hilbich et al., 2011; Rist and Phillips, 2005; Zhou et al., 2015). Furthermore the systematic investigation of soil moisture along a large elevation gradient reaching to Alpine permafrost conditions is non-existent so far.

The project SOMOMOUNT (Soil moisture in mountainous terrain and its influence on the thermal regime in seasonal and permanently frozen terrains, see also Pellet et al., 2016), which started in 2013 and is funded by the Swiss National Science Foundation, has the main objective to fill this data gap. Combining resources from the Swiss permafrost monitoring network (PERMOS) and the Swiss Federal Office for Meteorology and Climatology (MeteoSwiss), this project also aims at quantifying the influence of soil moisture on the ground temperature regimes at different altitudes.

In this contribution we will present a detailed description of the SOMOMOUNT monitoring network's instrumentation, monitoring strategy and calibration procedure, and discuss the measurement accuracy. Additionally, the data collected during the three first years of the SOMOMOUNT project will be discussed regarding the importance of the different water related processes, which are dominant at the different elevation bands. Hereby the differential impact of a three-week lasting heat wave on the different sites during summer 2015 is highlighted.



## 2 Soil moisture network

The soil moisture network established within the framework of the SOMOMOUNT project is currently composed of six fully automatic soil moisture monitoring stations installed along an elevation gradient ranging from 1205 m.a.s.l. to 3410 m.a.s.l., which spans from the Jura Mountains to the western Swiss Alps (Fig. 1). It is designed to be compatible with the

existing low elevation soil moisture monitoring network SwissSMEX (Mittelbach and Seneviratne, 2012) as well as the stations of the Swiss Permafrost Monitoring Network (PERMOS). Finally the high elevation soil moisture and permafrost monitoring station of Cervinia, Italian Alps, (Pellet et al., 2016; Pogliotti et al., 2015) is also included in the comparative analyses.

### 2.1 Instruments

Three different types of sensors are used within the SOMOMOUNT network: the SMT100 (TRUEBNER GmbH, Germany) based on the frequency domain reflectometry (FDR) technique, the TRIME-PICO64 (IMKO GmbH, Germany) based on the time domain reflectometry (TDR) technique and the PR2/6 (Delta-T Device Ltd, UK) based on the capacitance technique. Both the SMT100 and the PICO64 sensors are measuring simultaneously soil moisture and ground temperature. The sensor characteristics are listed in Table 1.

Both FDR and TDR methods are indirect measurement techniques that use electromagnetic waves to estimate the dielectric permittivity of the ground and relate it to the soil volumetric water content (VWC). The SMT100 sensors are composed of a ring oscillator which feeds a 10cm long transmission line (Fig. 2). The sensors emit an electromagnetic wave at a fixed frequency along the line which interacts with the surrounding medium and, when buried in the ground, with the VWC: the higher the VWC, the higher the effective dielectric permittivity, leading to a lower wave propagation velocity and thus to a

lower oscillation frequency (Schlaeger et al., 2005). The SMT100 sensors are the newest generation of the so-called SISOMOP sensors, which have been used to monitor soil moisture at Schilthorn (one of the high elevation permafrost sites, see section 2.3) since 2007 (Hilbich et al. 2011), demonstrating the sensors robustness and capability to measure in mountainous areas. Furthermore, laboratory experiments performed by Mittelbach et al. (2012) showed that the SISOMOP sensors have a similar absolute accuracy (±3 vol.%) compared to three other, more widely used, FDR sensors.

The PICO64 sensors are based on the standard TDR technique, which relates the travel time of an electromagnetic wave to the dielectric permittivity of the medium surrounding the sensors, which can in turn be related to the VWC of said medium. The PICO64 sensors emit an electromagnetic impulse at a frequency of 1 GHz, which travels along two 16cm long parallel rods, is reflected at their ends and returns back to the sensor to be recorded. The travel times are then related to the VWC using a general calibration based on Topp's equation (Topp et al., 1980). The spacing between the two rods of the PICO64 is

40mm (Fig. 2), which results in a measurement volume of ~1.25L (~10cm diameter around the rods). This sensor was selected for its high absolute accuracy (±1 vol.%, IMKO, 2015) and because it corresponds to the new generation of TRIME-



EZ sensors used by the SwissSMEX monitoring network (Mittelbach et al., 2011). Additionally, due to its large measurement volume, this sensor is particularly suitable for heterogeneous media (IMKO, 2015).

Finally, the PR2/6 sensor is a 100cm long down-hole water content sensor measuring soil moisture at 6 different depths (10, 20, 30, 40, 60 and 100cm) using the capacitance technique (Fig. 2). This technique relies on the fact that the charging time of

an electromagnetic field depends on the capacitance of the soil, which in turn depends on the dielectric permittivity and thus can be related to the VWC of the surrounding soil. Each measurement depth comprises a pair of stainless steel rings, which transmit the 100 MHz electromagnetic signal into the ground, and one detector, which records the returned signal. The sensor is lodged in an access polycarbonate tube of 25mm diameter and its measurement volume is ~10cm diameter with an absolute accuracy of ±6 vol.% (Delta-T Device, 2008). This sensor was selected for its measurement depth and its easy

installation. However, it is not suited for heterogeneous subsurface or coarse grained material since a good contact between the access tube and the soil is necessary. Furthermore, at least 1m soil is needed for its installation, thus it was only used at Frétaz (Sect. 2.3).

## 2.2 Network design

Each soil moisture station is equipped with 4 to 6 sensors along a vertical profile. The standard instrumentation consists of

one SMT100 at 10cm, two at 30cm and one at 50cm as well as one PICO64 at 30cm and one at 50cm (Fig. 2). The doubled sensors at 30cm were installed to check for instrumental drift on the long term. Depending on the soil characteristics, the installation of the complete instrumentation at all depths was not possible at all sites. The site specific set-ups are summarized in Table 2.

The same sensor installation procedure was followed at all sites and is based on the criteria described in Krauss et al. (2010)

and Mittelbach et al. (2011). While digging the pit for the sensor installation, each soil horizon was stored separately in order to preserve and restore the initial soil profile. At the depth of each sensor, up to two soil samples were collected at the side for granulometric analysis and water content determination. The sensors were then installed in the undisturbed soil with the blade in vertical position to avoid ponding (Fig. 2 and Fig. 3). Finally, the soil was refilled according to the original order of horizons and compacted to restore its original density. Additionally, larger samples of soils (about 8 L) were collected in the

vicinity to perform material specific calibration of the sensors (Sect. 3).

The data are recorded using a CR1000 data logger (Campbell Scientific) and transmitted with wireless transfer to an ftp server. The measurement interval is depending on the electrical power capacity of each station. At Frétaz and Moléson, where direct connections to the power grid are available, a 10min interval was chosen to match the setup of the SwissSMEX network (Mittelbach et al., 2011). At all the remaining stations, solar panels are used for power consumption and thus a

30min measurement interval was selected, except at Dreveneuse were the shaded location requires an even longer interval (60min).





### 2.3 Field sites

The site selection for the installation of the long term soil moisture monitoring stations within the SOMOMOUNT project was constrained by the following criteria:

 i. high enough elevation, so that the ground thermal regime is affected by seasonally or permanently frozen
conditions.
 ii. equal distribution along an altitudinal gradient.
 iii. availability of additional meteorological data and if possible ground temperature data.
 iv. easy access on site and a minimum of 50cm of fine grained material, to guarantee the installation of the sensors.

The stations were installed in collaboration with the Swiss Federal Office for Meteorology and Climatology MeteoSwiss (cf.
SwissMetNet, http://www.meteoswiss.admin.ch) for stations at middle elevation and the Swiss Permafrost Monitoring Network PERMOS (http://www.permos.ch) for stations at higher elevation (see Table 2). Located in the western part of Switzerland, the SOMOMOUNT network covers an elevation range from 1205 to 3410 m.a.s.l. with altitudinal differences between stations of 400-500m (Fig. 1).

### 2.3.1 Frétaz (FRE)

Frétaz is the lowermost field site within the network with an altitude of 1205 m.a.s.l. It is located in the western part of Switzerland on the first crest of the Jura Mountains. For the reference period 1981 to 2010 the mean annual air temperature was 6°C and the annual sum of precipitations 1333 mm $y^{-1}$ (MeteoSwiss). The soil moisture monitoring station is installed within the perimeter of the weather station belonging to the MeteoSwiss automatic network SwissMetNet, where ground temperatures down to 1m depth were also measured until 2005.

During the monitoring period 2013-2016 no freezing of the ground was observed. However, during the period 1981-2005 several short occurrences of below freezing temperature were recorded at 5cm and 10cm (MeteoSwiss). The surface cover consists of managed grass following the general directives from MeteoSwiss (grass cover maintained at all times at a few cm). The soil is composed of a unique layer of sandy loam down to 50cm (Table 3). According to geophysical surveys (Electrical Resistivity Tomography, ERT) the limestone bedrock is located at 5 to 10m depth underneath the station (see
Pellet et al. 2016).

### 2.3.2 Dreveneuse (DRE)

The Dreveneuse field site is located at an altitude of 1650 m.a.s.l. in a small North orientated valley within the Swiss Pre-Alpine region, where the mean annual air temperature is around 5°C (Morard, 2011). The soil moisture station is installed on a vegetated talus slope near an automatic weather station, and ground temperatures are monitored in two boreholes down to 5
and 14m depth.





This site is situated below the lower altitudinal limit of permafrost occurrence in the Alps but is still affected by permafrost conditions due to complex air circulation within the talus slope (Delaloye, 2004). The coarse limestone blocks composing the talus slope are covered by a single layer of organic rich sandy loam (Table 3, Fig. 3b). The surface is covered by moss and spruces and according to repeated ERT soundings as well as drilling logs, the talus slope is approximatively 11m thick

(Morard, 2011).

### 2.3.3 Moléson (MLS)

The Moléson soil moisture station is situated at 1974 m.a.s.l. on top of the eponym mountain in the Swiss Pre-Alpine region. For the reference period 1981 to 2010 the mean annual air temperature was 3°C and the annual sum of precipitations is 929 mm y$^{-1}$ (MeteoSwiss). As for FRE, the soil moisture station is integrated within the perimeter of a SwissMetNet station.

The site is affected by seasonal freezing processes down to maximum 10cm for the monitoring period 2013-2016. As for FRE the surface cover consists of managed grass and no apparent layering was found in the soil profile (homogeneous layer of silty loam down to 50cm, see Fig. 3 and Table 3). According to ERT measurements and the construction journal of the weather station, the bedrock, which consists of limestone, is located at around 75cm depth underneath the station.

### 2.3.4 Gemmi (GFU)

The Gemmi soil moisture monitoring station is located at 2450 m.a.s.l. in a West-orientated valley within the main alpine ridge of Switzerland (Fig. 1). The site receives between 1800 and 2500 mm y$^{-1}$ of precipitations per year and has a mean annual air temperature around 0°C (Krummenacher et al., 2008). The monitoring station is installed on a solifluction lobe in the direct vicinity of a 1m deep temperature profile installed in 1988 (Krummenacher and Budminger, 1992) and a weather station installed during summer 1999.

This site is situated just below the lower limit of permafrost occurrence in the Alps and therefore undergoes marked seasonal freezing processes down to at least 1m depth. The surface is covered by grass during the summer and the uppermost 10cm of the ground is composed of an organic rich silty loam layer (Table 3, Fig. 3). According to ERT measurements performed in 2014, the bedrock is located at around 5m depth underneath the station (Pellet et al., 2016).

### 2.3.5 Schilthorn (SCH)

The Schilthorn field site is situated at an elevation of 2900 m.a.s.l. on a small plateau in the North-facing slope of the Schilthorn summit in the northern Swiss Alps (Fig. 1). The average annual sum of precipitation is around 2700 mm y$^{-1}$ (Imhof et al., 2000) and the mean annual air temperature is about -3°C (PERMOS, unpubl.). The soil moisture station is installed next to an automatic weather station and two boreholes (14m and 100m deep), where ground temperatures have been monitored since 1998 (Harris et al., 2001).

Permafrost was first discovered at Schilthorn in 1965 and has been extensively investigated since 1999. The depth of the seasonally unfrozen soil layer (the so-called active layer) can reach up to 10m (PERMOS, 2013). The surface cover is





vegetation free and consists of a layer of fine grained debris with material ranging from loamy sand to sand (Table 3) which reach several meter thickness according to ERT measurements (Hilbich et al., 2008). SCH is the only station of the SOMOMOUNT network where soil moisture has already been monitored since end of August 2007 (Hilbich et al. 2011).

### 2.3.6 Stockhorn (STO)

The highest soil moisture monitoring station of the SOMOMOUNT network is located at an elevation of 3410 m.a.s.l. on the Stockhorn plateau in the Western Swiss Alps. This East-West orientated mountain crest has a mean annual air temperature of about -5°C (PERMOS, unpubl.) and the annual precipitation sum is estimated to be around 1500 mm y$^{-1}$ (King, 1990). The soil moisture station is installed in the vicinity of an automatic weather station as well as two boreholes measuring ground temperatures since summer 2000 down to 17 and 100m depth (Harris et al., 2001).

The Stockhorn plateau is underlain by at least 100m deep permafrost, which is strongly affected by 3D topography effects of the site (Gruber et al., 2004). The active layer thickness can reach up to 5m (PERMOS, 2013). The surface is free of vegetation and consists of a 1m deep layer of fine grained debris ranging from sand to loamy sand (Table 3) underlain by Albit-Muskovit schist bedrock (Gruber et al., 2004).

### 2.3.7 Additional stations

In addition to the SOMOMOUNT network we used the stations of Sion (SIO, Mittelbach and Seneviratne, 2012) and Cervinia (CER, Pellet et al., 2016; Pogliotti et al., 2015) for comparative analysis. The first one is part of the SwissSMEX network and is located in the Rhone valley at an elevation of 490 m.a.s.l. Since 2009, soil moisture is measured down to 80cm depth within the perimeter of the SwissMetNet station. Conversely, Cervinia is a high elevation (3100 m.a.s.l.) permafrost monitoring site managed by the regional environmental protection agency of Val d'Aosta (ARPA). Since 2006,

the site is equipped with two boreholes (7m and 14m deep) as well as an automatic weather station and one soil moisture sensor at 20cm depth.

### 2.4 Data processing

To ensure the quality of the soil moisture and ground temperature data, two different automatic filters are applied: a technical filter and a temporal filter. The filters are based on guidelines from Dorigo et al. (2013).

The technical filter is designed to eliminate all unrealistic values due to technical issues. Firstly, a threshold method is applied to detect and remove measurements outside of the plausible ranges (<0% and >80% for VWC and <-20°C and > 30°C for ground temperature). The threshold for VWC used here was empirically determined based on the data from all SOMOMOUNT stations. It is slightly higher than the 60% proposed by Dorigo et al. (2013) for the International Soil Moisture Network. Secondly, the values collected with insufficient battery voltage (< 10 V) are removed, since too low

power supply can disturb the measurements. For the SMT100 sensors, readings with a raw sensor output (given in so-called moisture counts, *MC*, see Truebner, 2016) outside of the range defined in the laboratory ($MC_{water} \approx 9000$ and $MC_{air} \approx 20000$)



are also excluded. At all sites the technical filter eliminated less than 0.1% of the measured values except at GFU, where the PICO64 sensors had a default in wiring and thus 8.3% of the measured data were excluded.

The temporal filter is designed to eliminate any VWC value exhibiting unrealistically large temporal variability (random spikes). Three-day running means ($r_{mean}$) and standard deviations ($r_{stdev}$) are calculated for all sensors, and the values lying

outside the range defined by $r_{mean} \pm x \cdot r_{stdev}$ are removed, where $x$ is an empirically determined site specific tolerance factor (3 at FRE and GFU, 4 at DRE, SCH and STO and 5 at MLS). This filter is applied to soil moisture and ground temperature measurements with an elimination rate varying between 0% (DRE) and 0.3% (FRE and SCH) of the measured data.

Numerous data gaps occurred at the different SOMOMOUNT stations during the monitoring period 2013-2016 (see Fig. 6). They were mainly caused by problems related to power supply (large data gap for all the sensors at one station e.g. autumn

2013 at FRE and MLS), data logging (short gaps for all sensors of the station e.g. winter 2014 at GFU) or sensor malfunction (single sensors for variable time, e.g. PICO64 at 50cm end of summer 2016 at STO). Given the highly variable nature of soil moisture, no gap filling technique was applied.

## 2.5 Complementary analysis

All soil moisture datasets used in the following analysis were homogenised to hourly mean values. For the elevation

dependency investigation (Sect. 5.2), annual and seasonal means were calculated using the year 2015, since all stations except GFU have complete data series during that period. The data gaps at GFU (24.02-01.03.2015 and 24.04-25.05.2015) were filled for that analysis only by linear interpolation between the nearest available data points. Finally, for the analysis of the transport of moisture through the ground we used the so-called moisture orbits (see Sect. 5.1).

To analyse and understand in detail the temporal evolution of soil moisture, additional datasets such as weather and ground

temperature data are needed. The stations of SIO, FRE and MLS are located next to SwissMetNet stations and thus data sets with good quality are available. This is not always the case at the high altitude/permafrost stations DRE, GFU, SCH and STO, where high altitude related logistical problems in maintenance may lead to data gaps and precipitation is not measured at all.

For the stations without explicit precipitation measurements, precipitation data were extracted from the 2km gridded dataset

generated by MeteoSwiss (MeteoSwiss, 2014), which is based on 430 observation stations across Switzerland and available at daily resolution. The data gap of the measured in situ air temperature series were completed using the two step quantile mapping approach described in Rajczak et al. (2016). Finally the snow duration was extracted from the near surface ground surface temperature variability using the snow index method described in Staub and Delaloye (2016).




## 3 Calibration

In order to increase the accuracy of the absolute soil moisture measurements, the SMT100 sensors require a material specific calibration (Table 4). Using the large soil samples collected at each site, a material specific calibration was performed in the laboratory following the general procedure outlined by Starr and Paltineanu (2002).

In a first step, the entire soil sample was oven dried and packed into a plastic container at approximatively the field bulk density. A SMT100 sensor was then inserted in the sample and its raw outputs were continuously recorded. Finally, a soil sample was collected using a standard measurement cylinder. In a second step, 200ml of water were added to the calibration soil (increase VWC by about 3-5%) which was subsequently thoroughly mixed to get a homogenous repartition of the water. The SMT100 was then reinserted, its raw outputs recorded and a soil sample collected. This operation was repeated until

saturation of the soil material was reached, yielding 9 to 12 calibration data points depending on the soil type.

The volumetric water content (VWC, $\theta_v$) of the collected samples was determined following the standard gravimetric method. First the samples were weighted and oven dried at 105°C during 24 hours for the mineral soils and 60°C during 48 hours for organic soils. The dry samples were then weighted to determine the gravimetric water content (mass of water which evaporated $\theta_g$) and to calculate the dry bulk density ($\rho_b$). Finally, $\theta_v$ was obtained using Eq. 1:

$$\theta_v = \rho_b \cdot \theta_g \qquad\qquad (1)$$

The gravimetric method was also applied to the in situ soil samples collected during the sensor installation. The calculated VWC ($\theta_v$) values obtained from both the laboratory and the in situ samples were then fitted to the SMT100 raw data (Moisture Counts, *MC*) using a linear (Eq. 2) and an exponential relation (Eq. 3).

$$\theta_v = a \cdot MC + b \qquad\qquad (2)$$

$$\theta_v = c \cdot e^{(MC \cdot d)} \qquad\qquad (3)$$

Table 4 lists the value of the parameters *a*, *b*, *c* and *d* for each station as well as the number of samples considered and the goodness of the fit for both methods. At GFU, two different material specific calibrations for mineral and organic soil were realized due to the clear layering of the soil profile (Fig. 3). At all locations except DRE the linear relation yields higher $r^2$ and lower RMSE than the exponential one. For consistency, the linear calibration is used for all sites since it yields the best

fits in most cases and a still very satisfactory fit at DRE.

For the PICO64, the built-in calibration based on Topp's equation (Topp et al., 1980) was used and no additional material specific calibration was performed according to Mittelbach et al. (2011). In our study, the PICO64 is mainly used as reference sensor regarding future comparison of data between the SwissSMEX soil moisture network and the SOMOMOUNT network.

Similarly, the manufacturer's calibration was used for the PR2/6 sensor (Delta-T Device, 2008), which is mainly used for test purposes within the SOMOMOUNT network at the moment. Depending on the middle-term results further PR2/6 sensors might be added later to the network.





## 4 Results

### 4.1 Sensor comparison and consistency

At 30cm depth two SMT100 sensors were installed in parallel in order to investigate potential instrumental drift over longer time periods. Comparing their outputs also allows us to assess the quality of the sensor installation, the reliability of the
measurements and potential spatial heterogeneity.

At FRE, DRE, MLS and GFU the correlation between the VWC measured by the two sensors was found to be satisfactory (lowest correlation at MLS: $r^2 = 0.766$, Fig. 4). Deviations from the one-to-one correspondence of the two sensors (black line) can be attributed to time delays in infiltration events and/or evaporation events. This is particularly visible at MLS, where the increase of VWC is systematically recorded first by the left sensor (y axis) and later by the right sensor (x axis).

At DRE (Fig. 4b) the right sensor shows consistently higher values (~5-10 vol.%) than the left sensor, but the relation between the measured VWC values is almost constant. This is due to the soil composition (sandy loam rich in organic matter and with a very low bulk density), which has a high hydraulic conductivity leading to almost instantaneous increase in VWC following precipitation events.

DRE is also the site with the largest RMSE (0.566°C) between the measured temperatures at 30cm depth, indicating that
different physical processes may influence the two sensors (see Sect. 5.3). At FRE, MLS and GFU the measured temperatures correspond almost perfectly one-to-one showing no different physical processes. Therefore the VWC deviations are most likely due to soil heterogeneities.

Sensor comparison was only possible at four sites. At SCH the terrain prevented the installation of two sensors at 30 cm, whereas at STO two sensors were installed but only one gives reliable data. The second sensor at 30 cm depth at STO
probably suffers from bad coupling with the soil (air pockets near the blade and thus bad contact with the soil) stemming from a faulty installation due to the blocky subsurface.

The TDR-based PICO64 sensors, which have a higher absolute accuracy (Mittelbach et al., 2012) are also, are installed at 30cm depth. Similar to Fig. 4, the comparison between FDR and TDR sensors at the same depth (Fig. 5) shows a generally good correlation (lowest $r^2 = 0.541$ at STO) with some deviations from the one-to-one relation (black line). At FRE and GFU
the comparison between PICO64 and SMT100 soil moisture measurements yield very similar results to the comparison of the two SMT100 sensors, with slightly higher RMSE values. MLS shows a larger dynamic range and mostly higher values for the SMT100 sensor, but a similar temporal variability. At SCH and STO the differences between the sensors have a characteristic shape, but are centred on the one-to-one relation. It can be attributed to different onset of freezing and thawing processes at the two sensor locations (see also Fig. 6e-f). Additionally, a clear wet bias in the PICO64 measurements can be
observed at SCH.

Figure 5 only considers the sensors at 30cm (left) at all sites with the exception of SCH, where the 10cm sensor was used, since it is the depth of the only available PICO64. The same analysis was performed with the 30cm right and 50cm sensors



(Table 5). The overall results are very similar regarding statistical fit (lowest fit at MLS and highest fit at GFU) and observed patterns (not shown).

## 4.2 Soil moisture temporal evolution

Figure 6 shows the evolution of the measured VWC and ground temperature at all SOMOMOUNT stations from July 2013 until August 2016 for all sensor types. The temperatures exhibit a typical seasonal pattern with maximum values during the summer and minimum values in winter. The amplitude of these seasonal variations is specific to each site. DRE, GFU, SCH and STO show a clear drop of temperatures below the freezing point during the winter, whereas no freezing was recorded at FRE. At MLS negative soil temperatures were only observed at 10cm depth during 10 days in early winter 2016.

On the other hand, the VWC differs strongly at each station and no common pattern is found. The sites can be divided into two categories of soil moisture dynamics: A low elevation pattern at FRE and MLS characterized by a summer minimum of short duration and high VWC values for the rest of the year. The second category, typical for high elevations (SCH and STO), is defined by a long lasting VWC minimum in winter, and maximum absolute values accompanied by strong variability during the summer. GFU and DRE display features characteristic from both categories, namely a long winter minimum as well as VWC decrease during summer. At GFU the 10cm sensor is much more variable than the ones at 30 and 50cm and shows much higher values. This is due to the high organic content and high retention capacity of this particular soil layer (Fig.3d and Table 3).

Comparing the two summer seasons in 2014 and 2015 at FRE, MLS and GFU one can observed a stronger VWC decrease in 2015 at all sites. This marked soil moisture decrease is due to the exceptionally high air temperatures recorded in July 2015 (MeteoSwiss, 2016; Scherrer et al., 2016) leading to increased evaporation. However, the effect of this anomalous event on soil moisture is different at all sites. At MLS the effect is the most pronounced (44 vol.% VWC loss at 30cm) and the VWC in the uppermost layers still did not returned to their original values in May 2016. At FRE the effect is less marked (18% vol.% VWC loss at 30cm) and shorter but it can be observed down to 100cm. At DRE it is seen at all depths with a similar amplitude (12 vol.% VWC loss at 30cm), whereas at GFU the effect was strong but of short duration at 10cm (40 vol.% VWC loss) but almost not seen below. Finally, at the two highest stations (SCH and STO) no characteristic VWC decrease was observed.

To characterize the two patterns of soil moisture dynamics identified above in more detail and to analyse the processes controlling them, we focus on a 5 months period from spring to summer 2015 at the lowest and highest field sites, FRE and STO (Fig. 7).

At FRE the minimum VWC is reached during the summer, when air temperatures are highest and thus evaporation is maximal. No clear VWC maximum can be identified throughout the year (Fig. 6a) but multiple maxima are observed following precipitation events. The snow cover, which disappeared mid-March in 2015, reduces the link between VWC variations and the atmospheric conditions. During the snow melt period only a small VWC increase is seen, which could be attributed to conditions close to saturation throughout the winter. After the disappearance of the snow cover the variability of





VWC increases at all depths and is systematically related to precipitation events (VWC increase) and dry spells, both, with and without air temperature increases (VWC decrease). As expected, the uppermost layer (10cm) reacts stronger than the lower ones to atmospheric forcing, however, the response time is very fast and in some cases almost simultaneous at all depths.

At STO the evolution is very different. Minimum VWC values are recorded in winter, when the ground is entirely frozen and the maximum is reached in early summer due to the combined effect of snow melt and thawing of the ground (Fig. 6f and Fig. 7b). The latter is more important, since the VWC increase is observed only once the ground is entirely unfrozen, which in 2015 happened several days after the total snow melt out. With ground temperature below the freezing point, the snow meltwater is either running off directly at the surface or refreezes at the top of the frozen layer. In contrast to FRE (Fig. 7a)

the evolution of VWC at STO is mainly driven by ground temperatures and the snow conditions and is less affected by liquid precipitation. Three main stages of VWC evolution and ground thermal regime can be identified. The frozen stage is characterized by the lowest VWC values due to the frozen state of the ground and the insulating snow cover. It is followed by the so-called zero-curtain period, defined by Outcalt et al., (1990) as extended period of time with near 0°C temperature induced by latent heat effects in a thawing or refreezing active layer. During this period VWC increases/decreases slowly but

remains decoupled from precipitation events. Punctual lateral inflow and/or snow meltwater infiltration are also possible. Finally, the unfrozen stage coincides with the snow-free period and is characterized by high VWC variability coupled with the precipitation events. The same typical stages of VWC evolution have been described for different mountain permafrost sites (e.g. Hilbich et al., 2011; Pellet et al., 2016) and landforms (e.g. Zhou et al., 2015).

**4.3 Soil moisture spatial distribution**

In addition to its pronounced temporal variability, soil moisture is also spatially highly variable (Mittelbach et al., 2012). This is especially true in heterogeneous terrain with strong microtopographic differences (Brocca et al., 2007; Williams et al., 2009). At SCH and STO, where deep ground temperatures are recorded in two boreholes at close distance respectively, long-term monitoring shows significant ground temperature offsets within short distances (0.5K at SCH and 1K at STO even at 15m depth, PERMOS, 2013). These variations are partly due to spatial differences of VWC and ice content in the ground

(Hilbich et al., 2011, Python, 2015). To analyse the effect of near-surface spatial soil moisture variability on the thermal regime at both sites, three additional spatially distributed SMT100 sensors were installed at 30cm depth within a radius of 20m around the standard SOMOMOUNT vertical profile. At both sites the standard SOMOMOUNT profile ("profile" in Fig. 8) is located in the direct vicinity of the shallower borehole (13m deep at SCH, 17m deep at STO) and the sensors named "bh100" in Fig. 8 were installed near the 100m deep boreholes.

Figure 8 shows that all sensors undergo the typical high altitude evolution of VWC described above for STO: (i) low variation and minimum values during the frozen winter period followed by (ii) a slow increase and punctual infiltration events during the zero-curtain period in late spring and finally (iii) high variability and maximum values in summer. The zero-curtain period occurs twice during year: in late spring associated with ground thawing and in autumn associated with





ground freezing, and it exhibits different soil moisture behaviour depending on the season. The punctual infiltration events are only observed during the spring zero-curtain and can be explained by the occurrence of melt water infiltration through the snow cover (Scherler et al., 2010, Hilbich et al., 2011). These events are restricted to single sensors or to specific depths (Fig. 7b and Fig. 8) indicating a strong influence of the microtopography and the three-dimensional effects (slope,

preferential flow paths, spatial snow melt patterns etc.) on the soil moisture regime.

Main differences between the spatially distributed sensors can be observed regarding the timing and duration of the three stages, but also regarding the measured absolute VWC values. At SCH the maximum spread of measured VWC during summer is about 24 vol.% and about 17 vol.% during winter, whereas at STO the spread is clearly smaller (15 vol.% and 4 vol.%, during the respective seasons). At SCH the wettest sensor ("profile") is the one of the standard SOMOMOUNT

profile, which is located in the vicinity of the borehole which shows higher temperatures, whereas the driest sensor ("bh100") is located near the borehole with the lower temperatures.

The timing of the different stages between the four sensors can vary up to 56 days at SCH and 35 days at STO. However, the duration of each stage is consistent within each field site. At STO, the zero-curtain lasts systematically longer, whereas at SCH the frozen stage is longer, indicating higher ice and water content in the near surface at STO. This is in good agreement

with the measured temperatures, which are lower at STO than at SCH during the frozen stage.

## 5 Discussion

### 5.1 Dominant processes

To visualize how the moisture is transported through the ground we adapted the so-called thermal orbits (Beltrami, 1996) to soil moisture (moisture orbits). It consists of a scatter plot of simultaneously measured VWC at two different depths. The

shape of the resulting point cloud depends on the nature and speed of the water transfer processes, as well as on soil properties such as hydraulic conductivity, degree of saturation and porosity. Using moisture orbits of different time scales (annual, pluri-annual and daily), allows us to analyse the dominant processes playing a role in the temporal evolution of soil moisture.

### 5.1.1 Seasonal variations

To investigate the seasonal dynamic of soil moisture, we used the moisture orbits between 10 and 50 cm depth for the year 2015 (Fig. 9). Again, the field sites can be divided into two categories of soil moisture dynamics, controlled by different processes.

FRE and MLS exhibit similar moisture orbit shapes driven by precipitation events and evaporation. They are divided into three main stages. From January to May the VWC is maximal at both sensors, with some small variations due to snow melt

and/or precipitation events. Starting in June, the summer evaporation causes the VWC to decrease strongly at 10cm and slightly at 50cm. Then VWC simultaneously decreases at both depths due to the increased evaporation generated by the July





2015 heat wave. Finally, VWC increases again, with different speed at each depth. The main difference between the two stations is that at MLS the orbit is not closed. At the end of 2015 there is about 30 vol.% VWC less than at the beginning of the year. It indicates that at MLS the near surface VWC did not recover from the increased evaporation generated by the heat wave in July 2015 (cf. Fig. 6). On the contrary, at FRE the VWC returned to its original value. This can be explained by two

main factors namely the precipitation regimes and the soil properties. MLS received less precipitation than FRE (1036 mm y$^{-1}$ and 1254 mm y$^{-1}$ respectively in 2015, MeteoSwiss), which hampered the full recovery of soil moisture conditions after the heat wave. Additionally, the absolute quantity of water lost at MLS was much larger due to stronger evaporation. The amount of water available for evaporation is dependent on the soil properties. At MLS the soil type is silty loam and is able to retain more water than the sandy loam (soil type at FRE).

At SCH and STO the shape of the moisture orbit is controlled by freezing/thawing processes. It can be divided into 5 stages consistent with the frozen, zero-curtain and unfrozen state of the ground. At the beginning of the year both sensors show their lowest value (frozen stage). This is followed by a sharp increase at 50cm not seen at 10cm (1), followed by a strong increase at 10cm but not at 50cm (2) consistent with the melting of the ground from underneath, which takes place at different time at the two depths (spring zero-curtain). During the summer, the ground is unfrozen and strong variations are

recorded at both depths (3). The orbit is finally closed by a succession of VWC decrease at 10cm not seen at 50cm (4) and a strong decrease at 50cm but not 10cm (5), consistent with the downward propagation of the freezing front from the surface (autumn zero-curtain). The orbits are closed at both sites indicating no long-lasting perturbation during the year 2015 and similar winter conditions. These stages have been observed by e.g. Hilbich et al. (2011) and Zhou et al. (2015).

At DRE and GFU it is difficult to determine a clear temporal evolution of soil moisture. DRE exhibits a winter and summer

minimum (see also Fig. 6b) corresponding to the freezing of the ground and the summer evaporation peak respectively. This double minimum is also found at GFU but less marked especially at the deeper layer.

### 5.1.2 Soil type dependency

At DRE the shape of the moisture is almost diagonal indicating very rapid transfer of water from the surface downward and little storage in the uppermost soil layer (wet anomaly at 50cm). This is due to the soil properties. At DRE the complete soil

profile consists of organic rich sandy loam with very low bulk density (Table 3) underlain by large sized boulders. This soil type is characteristically highly draining (Beringer et al., 2001) and the water is rapidly transported through. Additionally the boulders underneath do not retain the water, likely preventing the creation of any shallow water table. Furthermore, and in contrast to the soil type below, the organic rich material at the surface retains the water and lead to enhanced summer evaporation and winter freezing.

At GFU the presence of an organic rich layer in the uppermost 10cm of the soil causes the measured VWC at 10cm to be highly variable and higher than in the remaining soil column yielding an almost horizontal moisture orbit shape. At 30 and 50cm the soil is composed of loam and sandy loam with much larger bulk densities (Table 3). The organic rich layer has a lower thermal conductivity (Beringer et al., 2001) than the other soil types thus reducing the influence of air temperature





(evaporation and/or freezing) at larger depth. Furthermore the soil types at 30cm and 50cm have a lower hydraulic conductivity (Cosby et al., 1984), which also contributes to the lower soil moisture variability at these depths.

### 5.1.3 Climate dependency

As seen in Fig. 6, the annual soil moisture dynamics is strongly influenced by the variations in atmospheric conditions such as the extreme temperatures of July 2015. Thus, the shapes of the moisture orbits can be different for each year even though the same processes are dominant (evaporation or freezing). Comparing all years where measurements are available (Fig. 10), one can see that at low elevation (FRE), where the evolution of soil moisture is mainly controlled by evaporation and precipitation, the moisture orbits of the year 2014 and 2016 are very similar in shape and amplitude, whereas 2015 is marked by an exceptional decrease of VWC at both depths.

Conversely, at high elevation in permafrost terrain (SCH), 2015 is not particularly anomalous and the same patterns with comparable amplitudes are observed in all years available. This is to be expected since the freezing/thawing processes occur every year, with only slight variations regarding snow duration and timing.

Finally, at GFU, which is an intermediate site with similar characteristics to both FRE and SCH, two very different orbit shapes can be observed. In 2014, the ground froze down to 50cm producing a moisture orbit similar to SCH and STO characterized by large variations occurring at the two sensors. Conversely, in 2015 and 2016 only the 10cm layer froze, yielding horizontal shaped moisture orbits.

### 5.1.4 Infiltration events

Moisture orbits can also be used to characterize single infiltration events and investigate further the influence of the soil type on the short term soil moisture evolution. We selected one precipitation event recorded at all six stations and plotted the moisture orbits for a period of 5 days (Fig. 11). At all sites except STO, this precipitation event yields clear moisture orbits of different shapes and amplitudes. The slope of the orbits indicates at which depth the VWC is most affected by the precipitation event and the amplitude the amount of infiltrating water.

At FRE, GFU and SCH the orbits are horizontal (SCH) to slightly inclined (FRE and GFU), showing that the strongest variations of VWC are occurring at 10cm. The wetting and drying phases are faster and the perturbation larger at 10cm than at 50cm. At SCH the maximum VWC at 10cm is reached after two hours while no variation is recorded at 50cm. This pattern is typical for highly draining soils such as sand (found at SCH). At FRE and GFU the orbits correspond to soils with smaller hydraulic conductivity and higher retention capacity (sandy loam and loam-sandy loam respectively, Table 3).

At DRE the moisture orbit has almost similar amplitude at both depths and the slope is about 45°. This is in good agreement with the annual moisture orbit described above. It indicates a rapid transfer of water through the soil and no storage at 10cm typical for the particular soil composition found at DRE.

At MLS the moisture orbit is almost vertical due to VWC increase/decrease at 50cm depth only, which indicates an instantaneous transfer of water through the uppermost layer. It is the station where the event was the smallest (+8 mm d$^{-1}$)





but the resulting perturbation was highest (> +7 vol.%). From Fig. 6c, it can be seen that, at the time of the precipitation event, the VWC at 10cm and 30cm depths were unusually low, enabling the water to pass through easily. The degree of saturation of the soil layer is thus another key factor influencing the soil moisture dynamics.

Our interpretation of the moisture orbit shapes accounts only for vertical transfer of water in the soil. However, lateral flows

can also play an important role. At DRE, MLS, SCH and STO the stations are located on slightly inclined slopes or at their bottom. Furthermore, permanent snow patches have been observed at SCH and STO on several occasions and may constitute a continuous water supply during summer (Python, 2015; Wicki, 2015). An example of these lateral processes can be seen at STO, where the precipitation event shown in Fig. 11e did not yield a clear moisture orbit. Its effect is lost in the daily moisture orbit patterns due to snow melt cycles. For each day shown in Fig. 11e, an oval shaped moisture orbit can be seen.

These daily cycle orbits are very similar in amplitude and structure. The uppermost sensor reacts first (wetting and drying) and the maximum VWC increase happens at the same time at both depths.

As seen above for MLS, the influence of a single precipitation event on soil moisture not only depends on the soil properties but also on the moisture conditions prior to the event. To investigate this process we computed the moisture orbits of three selected precipitation events at FRE and DRE, which were preceded by different soil moisture conditions (Fig. 12). The first

event in mid-May is a combination of low precipitation amount preceded by comparatively high VWC . The second event is of very different amplitude at both sites (+39 mm d$^{-1}$ at FRE and +10 mm d$^{-1}$ at DRE) but it marked the end of the summer 2015 heat wave at both sites. The last event is the same as in Fig. 11. It consists of a large amount of precipitation preceded by relatively high VWC.

At FRE, the first and third events yield similar moisture orbit shapes with a larger amplitude for the larger precipitation

event. However, the second event produces a diagonal moisture orbit with almost no VWC decrease. As for MLS above, the VWC is low at all depths enabling the precipitations to infiltrate quickly down to the deepest layers. The same is seen at DRE, where the second precipitation event is comparatively small (+10 mm d$^{-1}$) but yields the strongest and fastest VWC increase at depth

## 5.2 Altitude dependency

As seen above four main processes are driving the annual soil moisture dynamics, namely evaporation/infiltration and freezing/thawing of the ground. The respective predominance of one of these processes is dependent on the station location and more specifically on its elevation. Using all SOMOMOUNT stations as well as the SwissSMEX station of Sion (SIO) we investigated the elevation dependency of mean annual and mean seasonal VWC for the year 2015 (Fig. 13a).

The relation between soil moisture and elevation is clearly non-linear. Disregarding DRE (see Sect. 5.3), a distinct pattern

emerges. The mean annual VWC regularly increases with elevation until about 2000 m.a.s.l. and then decreases with increasing elevation. This tipping point corresponds also to a clear shift in the soil moisture regime. Below 2000 m.a.s.l. the maximum VWC is recorded in winter and the minimum in summer, whereas above this threshold the inverse occurs (maximum VWC in summer and minimum in winter).





This shift in soil moisture regime can be related to a series of variables, which are known to be important for mountain climates and which were also plotted against elevation (Fig. 13b-e). Air temperature (Fig. 13b), resulting from the radiation balance, controls the energy available for evaporation and freezing, whereas the precipitation amount (Fig. 13c) controls the water input at the surface. The snow cover duration (Fig. 13e) has several effects on the soil moisture dynamics: it insulates

the ground from the cold winter temperatures (yielding positive surface offsets, Fig. 13b) and it acts as a water retention layer, which stores water throughout winter and liberates it in spring/early summer.

From Fig. 13 the following relationships can be determined: ground- and air temperature, as well as the thawing degree days (absolute sum of positive temperatures per year) all linearly decrease with elevation and conversely the freezing degree days (absolute sum of negative temperatures per year), surface offset and the snow cover duration increase with elevation. There

is no clear trend in the precipitation distribution due to the strong microclimatic effects, however on larger spatial scale (continental) precipitation is known to increase with elevation (Smith, 1979).

At lower elevation (below 2000 m.a.s.l.), evaporation dominates the soil moisture regime, causing the summer VWC minimum. With increasing elevation air temperature decreases, precipitation increases and the snow cover duration is prolonged, explaining the increase of mean annual VWC.

At higher elevation (above 2000 m.a.s.l.), the ground thermal regime and more specifically the soil freezing process drives the soil moisture regime, causing the VWC minimum to shift from summer to winter, when freezing occurs. This is confirmed by the observed negative ground temperatures as well as the increasing freezing degree days. With increasing elevation, air and ground temperatures decrease yielding increasingly long duration of seasonally frozen ground and thus explaining the decreasing trend of VWC.

**5.3 Special case: Dreveneuse**

To summarise the findings from the SOMOMOUNT network presented above, a simple theoretical model of the evolution of soil moisture and its contributing factors with elevation can be visualised with the grey shading in Figure 14. Comparing the observations qualitatively with this model (circles in Fig. 14), it can be seen that DRE does not fit the model. The recorded mean annual VWC values are much lower than expected.

The case of DRE is particular not only for its soil moisture dynamics but also in terms of snow duration (longer than expected) and mean annual ground temperature (lower than expected). Both anomalies are due to site specific characteristics, which are independent from elevation.

The low mean annual ground temperature results from a complex air circulation within the underlying talus slope (Delaloye, 2004; Morard, 2011), which is made possible by the large interconnected pore space between the coarse blocks of the talus.

During winter, ascending warm air within the talus slope leads to cold air inflow at the bottom of the talus slope, where the soil moisture station is located. This process efficiently cools the ground even when the snow cover is present and has been observed at many similar talus slopes in low and high mountain regions (e.g. Delaloye and Lambiel 2005; Gude et al. 2003; Kneisel et al. 2000; Sawada et al. 2003; Wakonigg, 1996). In summer the reverse process takes place with outflow of cold





air from the inside of the talus slope at the bottom by gravity. The longer snow duration is due to the spruces and low vegetation surrounding the station, which efficiently traps the snow for extended period of time. Additionally the cold ground temperatures help to conserve the snow cover for a longer period.

At DRE, the relatively low elevation of the station implies a high air temperature and thus more energy available for
evapotranspiration. Additionally, the presence of vegetation at the surface induces water uptake. The reduced ground temperatures by the air circulation result in a negative surface offset (ground temperature lower than air temperature) and slightly positive freezing degree days, both indicating a seasonal freezing of the ground. Thus two phases of minimal VWC values are observed in summer and winter (see Fig. 6b). Additionally, the coarse grained soil type at DRE has low water retention properties and induces a fast transport of water to the deeper layers, affecting strongly the short term soil moisture
dynamics.

Our elevation dependent model is an empirical model developed using seven stations. Although the stations are regularly distributed with elevation (~500m steps) the representativeness of these locations is hard to assess. Additional low elevation soil moisture monitoring stations are available within the SwissSMEX network and fit well in the presented model. Comparison with further low- to middle altitude stations in the Black Forest region (Southwest Germany) show also good
agreement (Krauss et al. 2010). Finally, the high elevation monitoring station at Cervinia/Italy (3100 m.a.s.l., see Pellet et al., 2016) exhibits soil moisture dynamics comparable to STO and SCH and its mean annual VWC for 2015 fits well in the elevation model presented above.

However, the example of DRE shows that some of the processes playing a role for the soil moisture dynamics do not have a trivial elevation dependency. Precipitation in mountainous areas is especially difficult to monitor and the elevation factor is
hereby less important than topographic effects. As shown in Fig. 13c, the annual sum of precipitation at a given altitude can vary up to 2000 mm $y^{-1}$ depending on the location. From our field sites FRE appears to receive more precipitation than expected, whereas STO receives much less than predicted. Indeed, FRE is situated on top of the easternmost ridge of the Jura mountain chain, which is known to have a comparatively high annual precipitation sum, as is also shown by the neighbouring SwissMetNet stations of Chasseral (1599 m.a.s.l.) and Chaumont (1136 m.a.s.l.), which recorded annual
precipitation sums of 1509 mm $y^{-1}$ and 1240 mm $y^{-1}$ respectively for the period 1961-1990 (MeteoSwiss). Precipitation maxima in middle mountain ranges are often found at the highest elevations, whereas the precipitation distribution in high mountain ranges is strongly influenced by the prevailing wind directions and found along the windward slopes, with smaller values on mountain tops and in the lee of the mountains (e.g. Smith 1979). In addition, STO is located in the central alpine region, which is very dry due to the wind shading effects from the surrounding mountain crests. This effect is also seen at the
station of Findelen (VSFIN), which is located about 3km from STO and which is also much drier than the stations at comparable elevation (Fig. 13c).





## 6 Conclusion

In this paper we presented a detailed description of the new soil moisture monitoring network for middle and high altitudes in Switzerland (SOMOMOUNT). Starting in summer 2013, six automatic stations have been set up along an elevation gradient ranging from 1205 to 3410 m.a.s.l.

The use of two types of standard soil moisture sensors for application in coarse grained terrain undergoing freeze/thaw cycles at middle and high elevation was shown to be reliable. Although the sensors are not specifically designed for freezing conditions, we found the measurements to be consistent, both regarding inter-sensor comparisons as well as in comparison with related variables such as ground temperature and precipitation. A standard calibration approach combining in situ and laboratory analysis was applied to improve the measurement accuracy. However the absolute value during the frozen period

remains difficult to assess even though both the SMT100 and the PICO64 sensors yield similar absolute values (±3 vol.% range). The measurements also confirmed that unfrozen water content is present at temperatures below the freezing point and that it can be measured with the sensors.

The data collected during the first three years of the SOMOMOUNT network revealed very distinct soil moisture dynamics at the different sites, which could be summarized into a simple elevation dependent model. At middle and low elevation,

annual soil moisture dynamics are controlled by evapotranspiration and precipitation events whereas at high elevation the freeze-thaw cycle is the main driving factor.

Marked inter-annual variations have been observed. However, depending on the site-specific properties, the impacts have been more or less important. The exceptionally high air temperatures of July 2015 induced a stronger and longer lasting soil moisture decrease than 2014 or 2016, but only for low- and middle-altitude stations. At high elevation (>2900 m.a.s.l.) no

effect of the 2015 heat wave was observed, since the soil moisture dynamics is predominantly controlled by the ground thermal regime.

Among the six soil moisture stations of SOMOMOUNT, and also in comparison with additional stations from other networks, the station of Dreveneuse is a clear exception to the elevation dependent theoretical model. This middle elevation site undergoes strong winter freezing as well as marked summer evaporation, the latter being due to the vegetation cover.

Due to complex air circulation within the underlying talus slope the ground temperatures are unusually low for this elevation. In addition, the soil properties favoured rapid water transport through the ground. The soil properties were found to play an important role in the short term soil moisture variations as well as in the mitigation or intensification of the extreme events.

**Competing interests**

The authors declare that they have no conflict of interest.





## Acknowledgments

This study was conducted within the SOMOMOUNT project funded by the Swiss National Science Foundation (project n$^o$ 143325). All data from the SwissMetNet weather stations have been provided by MeteoSwiss, the Swiss Federal Office of Meteorology and Climatology. We are thankful to the Land-Climate Dynamics group of S.I. Seneviratne, Institute for Atmospheric and Climate Science (IAC), ETH Zurich for providing the data from the SwissSMEX network (http://www.iac.ethz.ch/group/land-climate-dynamics/research/swissmex.html). We also thank our colleagues from ARPA for supplying the data relative to the Cervinia field site and the PERMOS network for the weather and temperature data at Dreveneuse, Gemmi, Schilthorn and Stockhorn.

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




**Fig. 1: Map of Switzerland showing the location of the soil moisture monitoring stations from the SOMOMOUNT and SwissSMEX project (Mittelbach and Seneviratne, 2012) as well as the ARPA monitoring station Cervinia (Pogliotti et al. 2015). Selected mountain weather stations form the SwissMetNet network used in Sect. 5.2 are also displayed.**

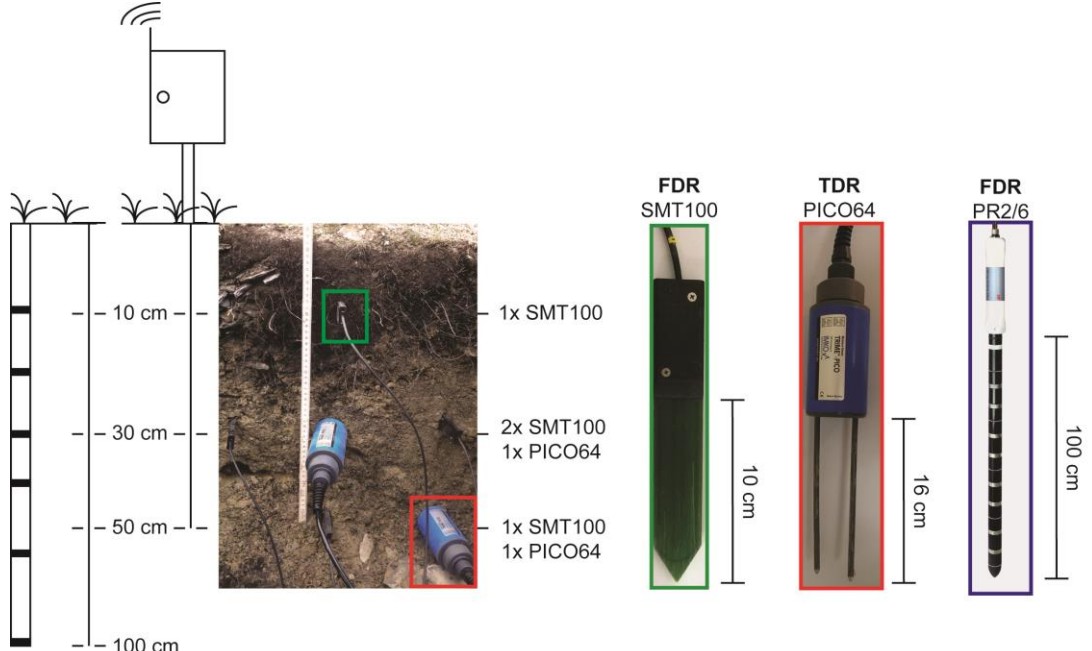

**Fig. 2: Instrumentation of the standard SOMOMOUNT station.**

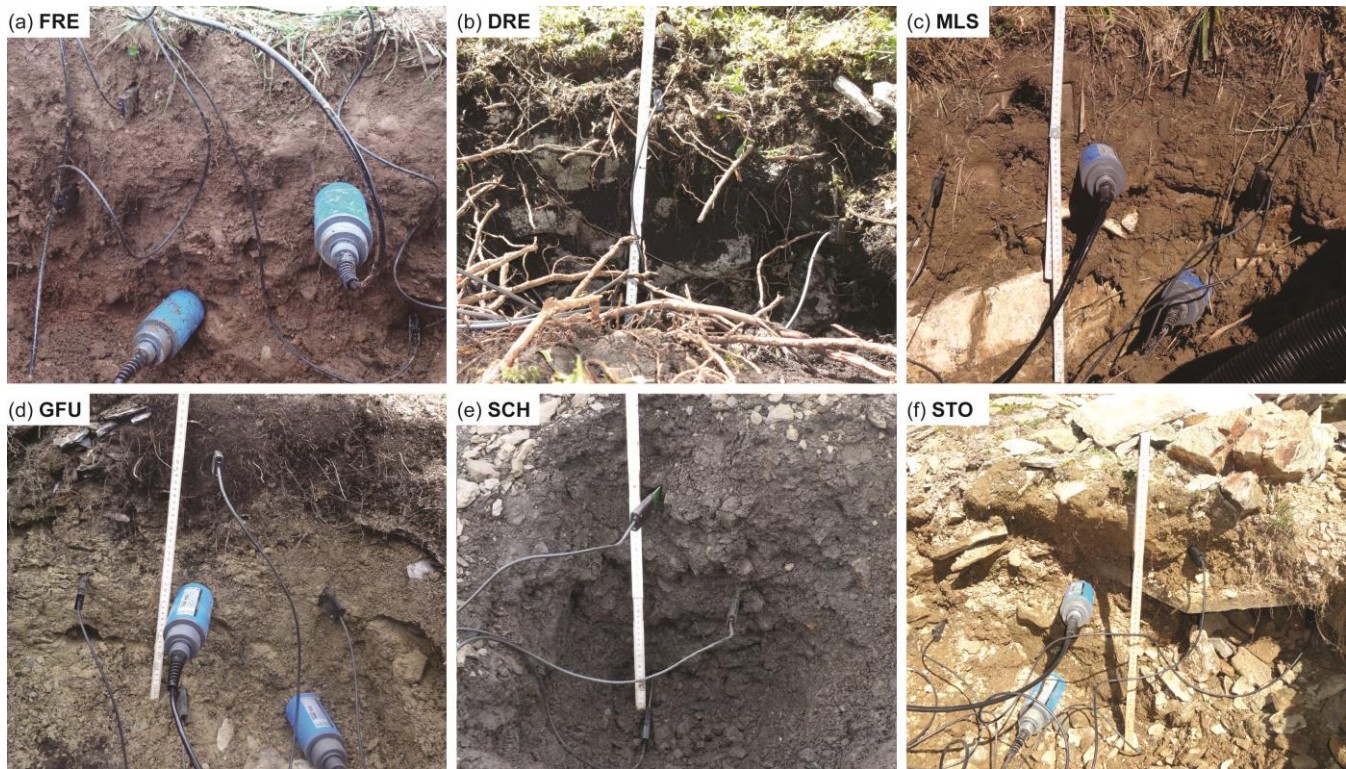

**Fig. 3: Illustration of the soil characteristics and sensor installation for all SOMOMOUNT stations.**




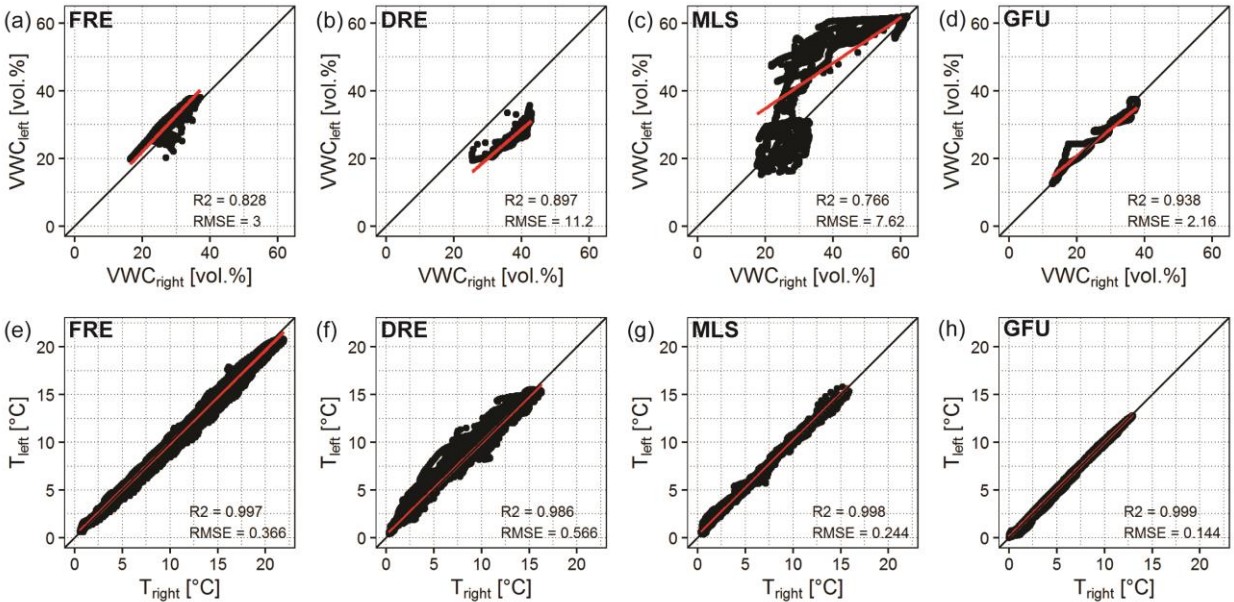

**Fig. 4: Comparison of measured soil moisture (upper row) and ground temperature (lower row) from both FDR measurements at 30 cm depth using the linear calibration at FRE (a, e), DRE (b, f), MLS (c, g) and GFU (d, h).**

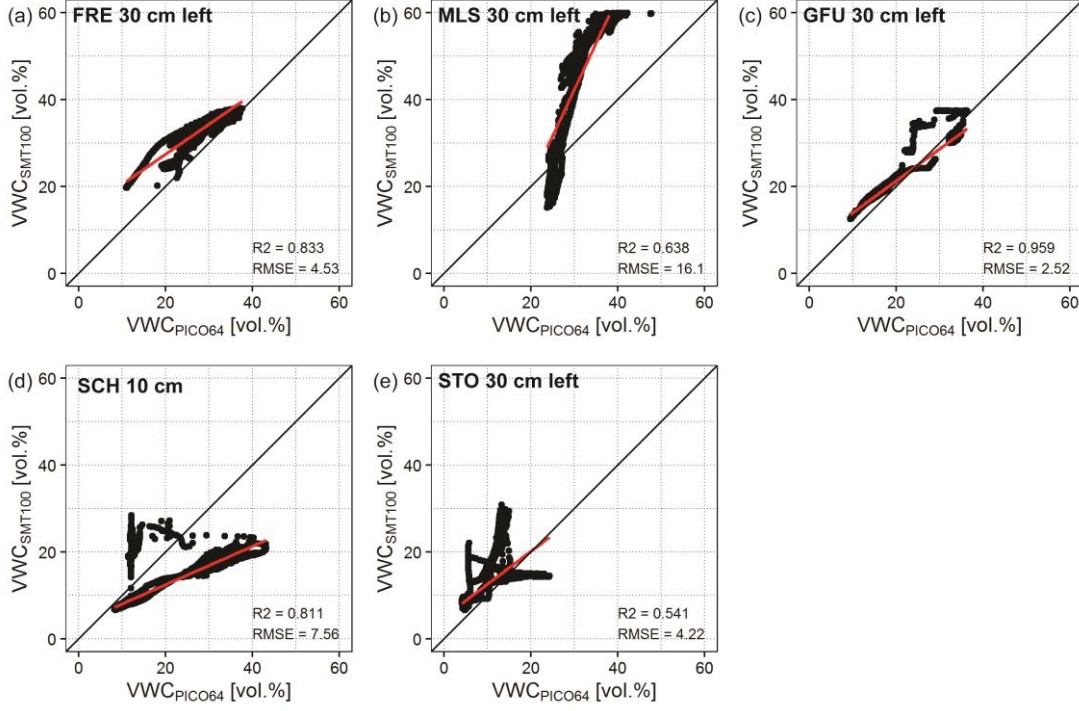

5   **Fig. 5: Comparison between TDR- (x-axis) and FDR-measured VWC (y-axis) at all sites. The linear relation is used for the FDR calibration.**





**Fig. 6: Measured VWC (upper panel) and ground temperatures (lower panel) at each SOMOMOUNT station (a-f). At FRE the uppermost panel displays the PR2/6 measured VWC. The vertical dotted lines at FRE and STO indicate the period analysed in Fig. 7.**





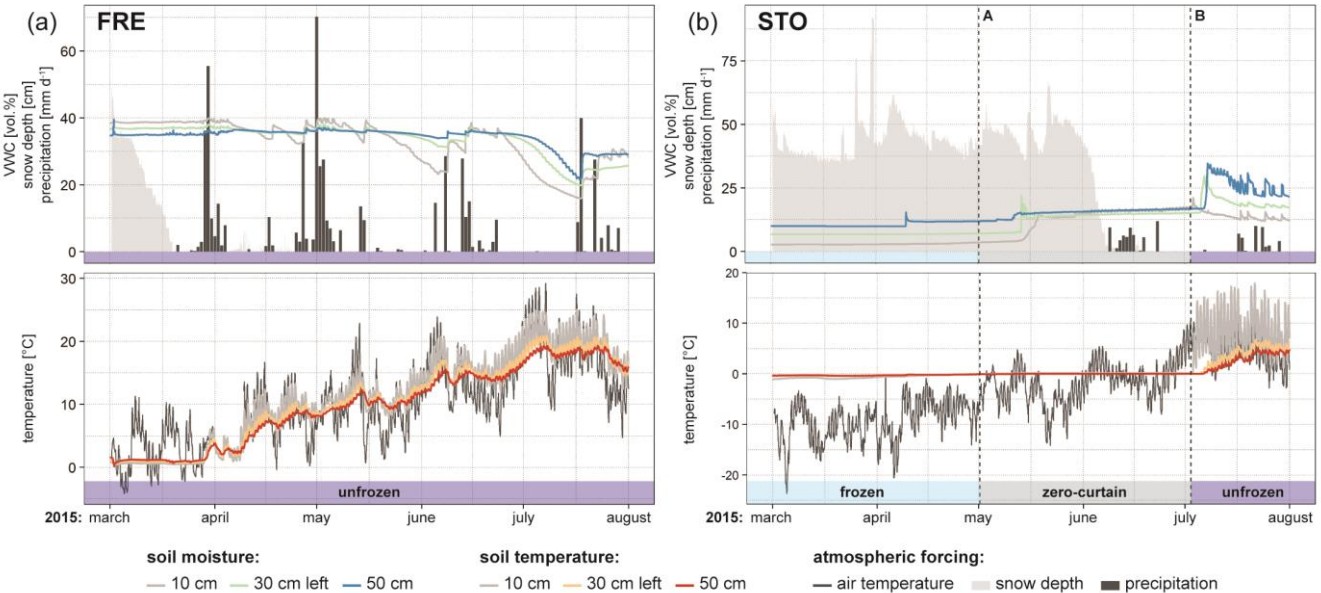

**Fig. 7: Measured VWC and ground temperature at FRE (a) and STO (b) from March to August 2015. In addition, daily air temperature, snow depth and precipitation sums are shown as well as the date of the transition between the different stages in the thermal evolution at STO at 10cm (dashed lines A and B, see text for details).**

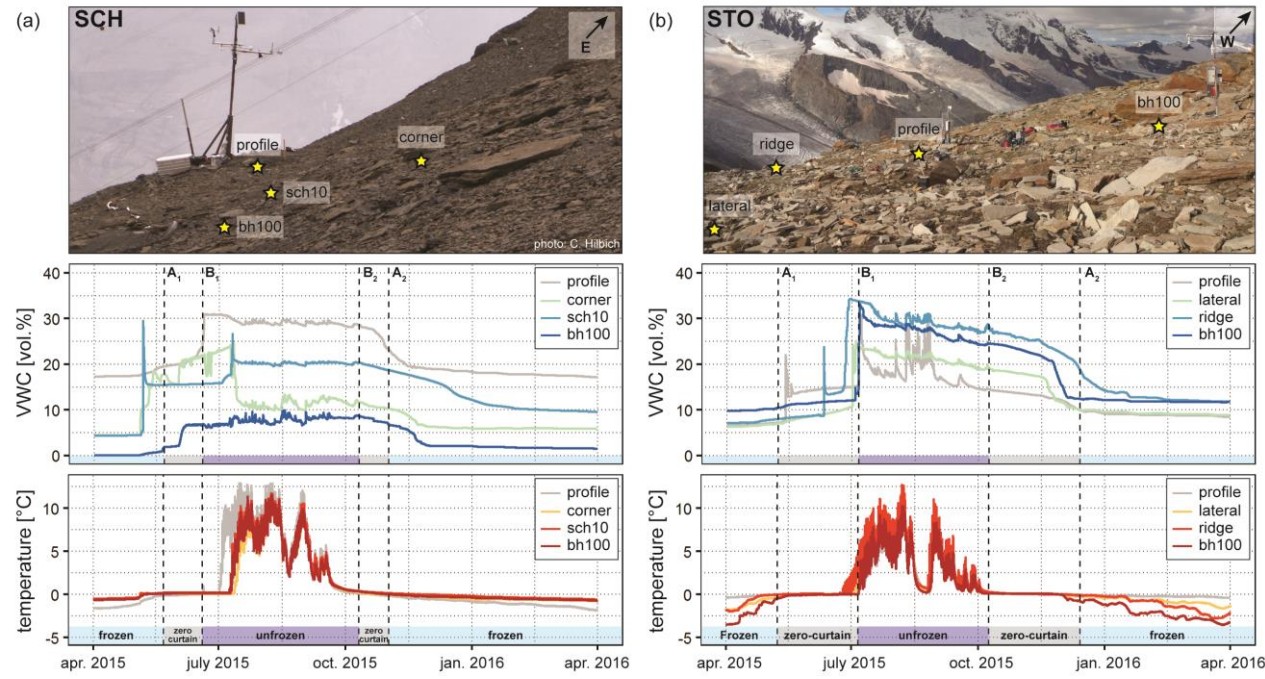

**Fig. 8: Location of the spatially distributed SMT100 sensors (upper panel) and measured VWC (middle panel) and ground temperature values (lower panel) at 30cm depth at SCH (a) and STO (b) between April 2015 and April 2016. The transitions between the typical stages are indicated with vertical dashed lines.**





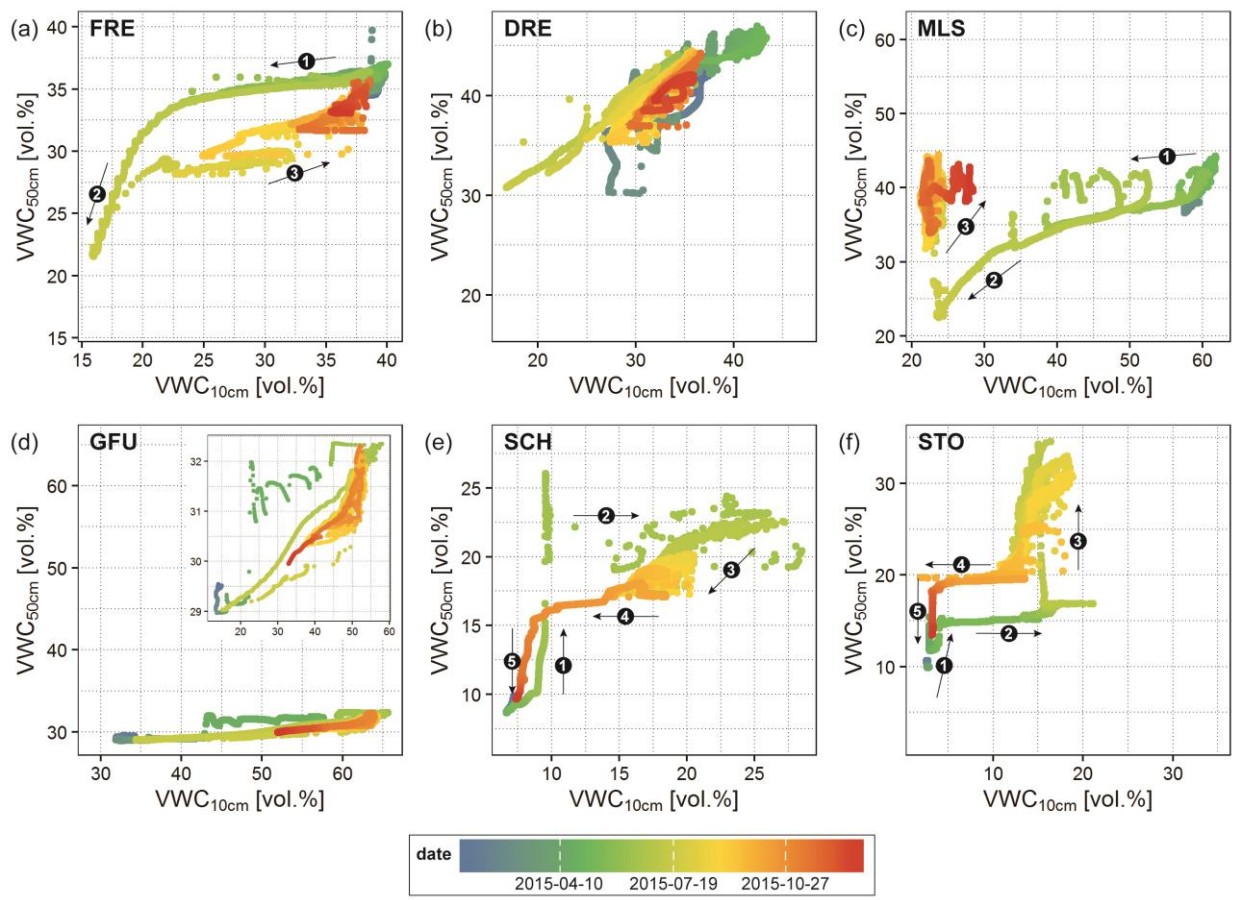

**Fig. 9: Moisture orbit at each SOMOMOUNT station from the 1st January to the 31st December 2015. The numbered arrows indicate the most important stages at each station as well as the sense of the evolution.**

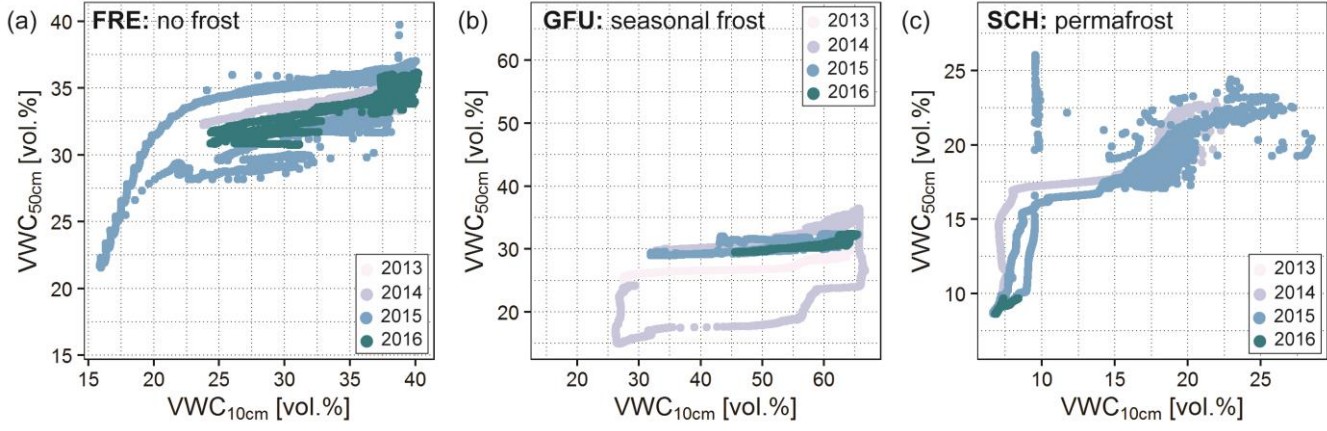

5    **Fig. 10: Moisture orbits at FRE (a), GFU (b) and SCH (c) for the consecutive monitoring years (Jan. 2014-Aug. 2016 at FRE, Aug. 2013-Aug. 2016 at GFU and Aug. 2014-June 2016 at SCH).**







**Fig. 11: Moisture orbit at each SOMOMOUNT station for one precipitation event between the 23ʳᵈ and the 27ᵗʰ August 2015. The VWC values are given as hourly mean and expressed as the change of absolute value compared to the first measurement (23ʳᵈ August at 01:00). The daily precipitation sums recorded (FRE, DRE and MLS) and extrapolated (GFU, SCH and STO) for the 24ᵗʰ August are indicated.**





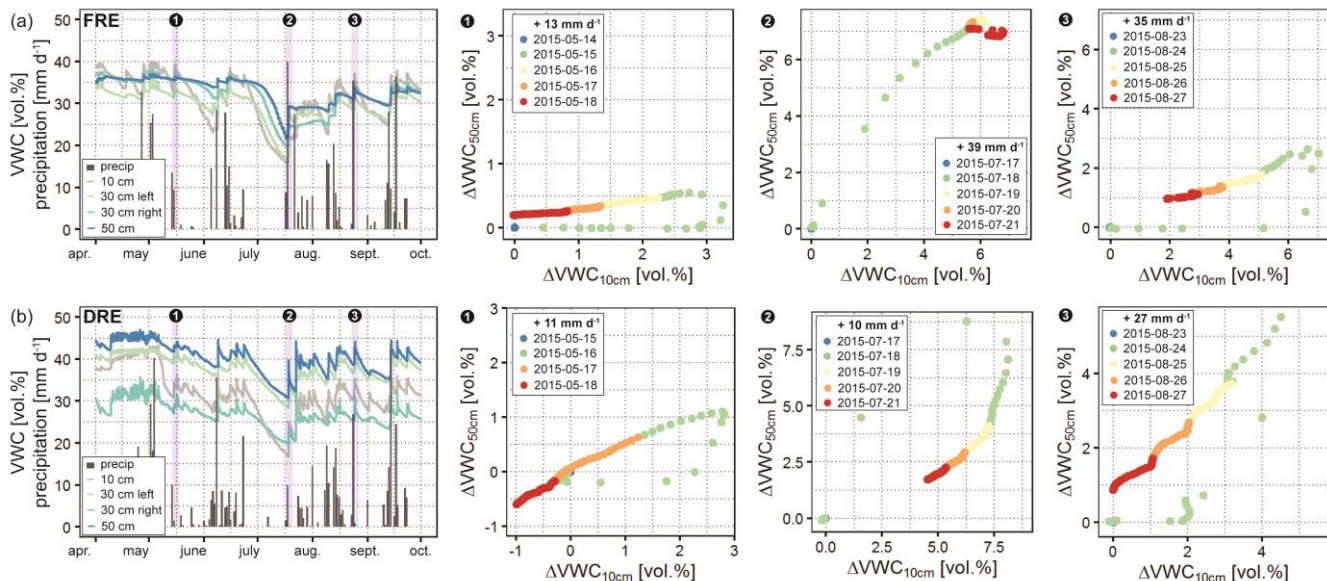

**Fig. 12: Moisture orbit at FRE (a) and DRE (b) for three precipitation events in 2015. The VWC values are hourly means and expressed as the change of absolute value compared to the first measurement (23$^{rd}$ August at 01:00). The daily precipitation sums recorded at the beginning of each event are indicated.**





**Fig. 13: Elevation dependency of the winter-, summer- and annual mean VWC at 30cm depth (a), air-, ground temperature and surface offset (ground minus air temperature) at 30cm (b), annual precipitation sum including selected SwissMetNet stations for comparison (cf. Fig. 1) (c), freezing and thawing degree days (calculated from ground temperatures at 10cm) (d) and snow duration (calculated from ground temperature at 10cm using the method described in Staub and Delaloye (2016)) (e) during the year 2015. The VWC values for SIO are part of the SwissSMEX network (Mittelbach and Seneviratne, 2012).**





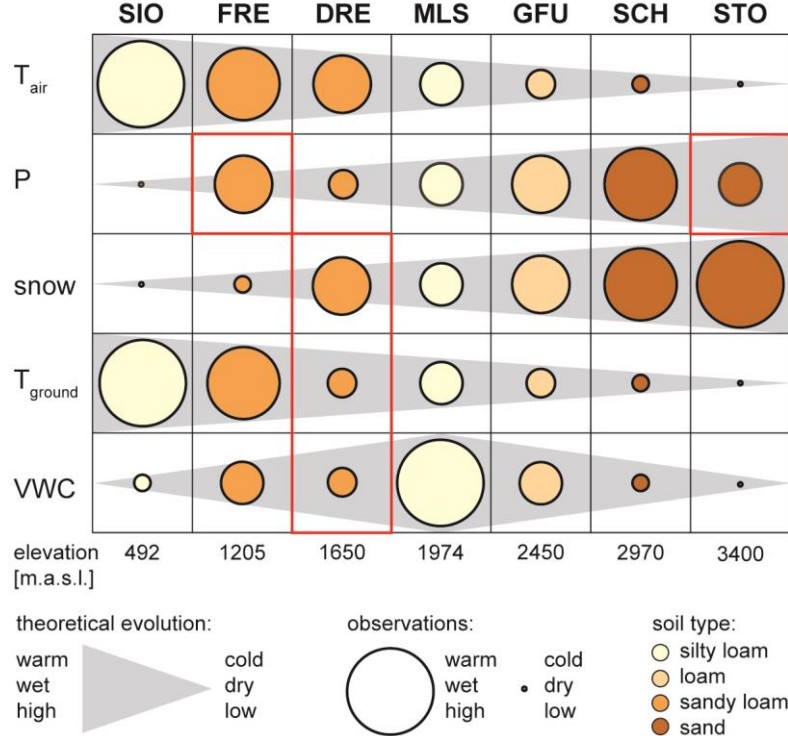

**Fig. 14: Conceptual model of the evolution of air temperature, precipitation, snow duration, ground temperature and soil moisture with elevation. The circles represent the observations from 2015 (see Fig. 13), the grey area the expected theoretical evolution and the colour scale the soil type. The mismatches between model and observations are highlighted in red.**

| Sensors | Measurement technique | range of VWC | operating temperature | Accuracy |
|---|---|---|---|---|
| SMT100 (Truebner GmbH, Germany) | FDR | 0 to 100 vol.% (60-100% limited accuracy) | -40° to 60°C | ±3 vol.% for 0-50 vol.% ±1 vol.% using medium specific calibration |
| PICO64 (IMKO GmbH, Germany) | TDR | 0 to 100 vol.% | -15° to 50°C | ±1 vol.% for 0-40 vol.% ±2 vol.% for 40-70 vol.% |
| PR2/6 (Delta-T Devices Ltd, UK) | capacitance | 0 to 100 vol.% | -20° to 60°C | ±6 vol.% for 0-40 vol.% ±4 vol.% using medium specific calibration |

5 **Table 1: Characteristics of the three types of soil moisture sensors used in the SOMOMOUNT network. All values in the table are provided by the manufacturers (Delta-T Device, 2008; IMKO, 2015; Truebner, 2016).**



| Site | Elevation [m.a.s.l.] | Sensor depth [cm] | | | Measurement interval | Start date | Related networks |
|------|------|------|------|------|------|------|------|
| | | SMT100 | PICO64 | PR2/6 | | | |
| FRE | 1205 | 10, 30, 30,50 | 30, 50 | 10,20,30, 40,60,100 | 10 min | 11.10.2013 | SwissMetNet |
| DRE | 1650 | 10, 30, 30,50 | - | - | 60 min | 26.09.2014 | PERMOS |
| MLS | 1974 | 10, 30, 30,50 | 30, 50 | - | 10 min | 17.10.2013 | SwissMetNet |
| GFU | 2450 | 10, 30, 30,50 | 30, 50 | - | 30 min | 17.07.2013 | PERMOS |
| SCH | 2970 | 10, 30, 50 | 10 | - | 30 min | 31.07.2014 | PERMOS |
| STO | 3410 | 10, 30, 30,50 | 30, 50 | - | 30 min | 27.08.2014 | PERMOS |

**Table 2: Summary of the station instrumentation and characteristics at the field sites Frétaz (FRE), Dreveneuse (DRE), Moléson (MLS), Gemmi (GFU), Schilthorn (SCH) and Stockhorn (STO).**

| Site | Depth [cm] | Particle size distribution [%] | | | Texture[a] | Bulk Density | Organic fraction [%] | Thermal regime |
|------|------|------|------|------|------|------|------|------|
| | | Clay [<2μm] | Silt [2-63μm] | Sand [>63μm] | | | | |
| FRE | 0-10 | 2.15 | 33.25 | 64.60 | Sandy loam | 0.95 | 0.1 | |
| | 10-30 | 2.51 | 36.09 | 61.40 | Sandy loam | 1.06 | - | No frost |
| | 30-50 | 3.03 | 30.77 | 66.20 | Sandy loam | 1.01 | - | |
| DRE | 0-50 | 0.60 | 24.40 | 75.00 | Sandy loam | 0.12 | 7.7 | Permafrost |
| MLS | 0-10 | 11.82 | 72.48 | 15.70 | Silty loam | 0.58 | 0.15 | Seasonal |
| | 10-30 | 15.14 | 77.56 | 7.30 | Silty loam | 0.71 | - | frost |
| GFU | 0-10 | 2.00 | 63.30 | 34.70 | Silty loam | 0.58 | 4.8 | Seasonal |
| | 10-30 | 2.15 | 39.65 | 43.20 | Loam | 1.52 | - | frost |
| | 30-50 | 2.60 | 46.50 | 50.90 | Sandy loam | 1.39 | - | |
| SCH | 0-10 | 1.00 | 14.49 | 84.50 | Loamy sand | 1.53 | - | Permafrost |
| | 10-30 | 0.59 | 8.71 | 90.70 | Sand | 1.35 | - | |
| STO | 0-10 | 0.26 | 6.44 | 93.30 | Sand | 1.42 | - | |
| | 10-30 | 0.59 | 20.41 | 79.00 | Loamy sand | 1.67 | - | Permafrost |
| | 30-50 | 0.72 | 21.58 | 77.70 | Loamy sand | 1.54 | - | |

[a]according to the USDA classification

**Table 3: Soil properties for each station.**





| Site | linear fit | | | | exponential fit | | | | n |
|------|-----------|-----|-----|----------------|-----|-----|-----|----------------|---|
|      | $a$ | $b$ | $r^2$ | RMSE [vol.%] | $c$ | $d$ | $r^2$ | RMSE [vol.%] | |
| FRE | -0.005630 | 98.15 | 0.96 | 2.76 | 707.1 | -0.0002680 | 0.93 | 3.66 | 16 |
| DRE | -0.006361 | 114.7 | 0.89 | 5.71 | 1219 | -0.0002889 | 0.96 | 3.51 | 12 |
| MLS | -0.008108 | 139.2 | 0.97 | 3.59 | 741.7 | -0.0002507 | 0.95 | 5.03 | 16 |
| $GFU_{min}$ | -0.005327 | 93.7 | 0.80 | 5.57 | 465.8 | -0.0002336 | 0.75 | 6.18 | 13 |
| $GFU_{org}$ | -0.007819 | 140.2 | 0.95 | 3.93 | 534.2 | -0.000209 | 0.93 | 4.83 | 14 |
| SCH | -0.004301 | 77.55 | 0.81 | 4.47 | 899.2 | -0.0002925 | 0.88 | 3.57 | 12 |
| STO | -0.004851 | 85.71 | 0.84 | 4.02 | 457.9 | -0.0002378 | 0.79 | 4.58 | 18 |

**Table 4: Parameters and statistics of the linear and exponential calibration curve for each station.**

| Site | 10 cm | | 30cm left | | 30cm right | | 50cm | |
|------|-------|--------------|-------|--------------|-------|--------------|-------|--------------|
|      | $r^2$ | RMSE [vol.%] | $r^2$ | RMSE [vol.%] | $r^2$ | RMSE [vol.%] | $r^2$ | RMSE [vol.%] |
| FRE | - | - | 0.833 | 4.53 | 0.934 | 2.12 | 0.504 | 10.4 |
| MLS | - | - | 0.638 | 16.1 | 0.848 | 12.1 | 0.553 | 10.1 |
| GFU | - | -- | 0.959 | 2.52 | 0.89 | 2.37 | 0.91 | 1.78 |
| SCH | 0.811 | 7.56 | - | - | - | - | - | - |
| STO | - | - | 0.541 | 4.22 | - | - | 0.807 | 6.93 |

**Table 5: Correlation ($R^2$) and RMSE between the TDR and FDR measured VWC at all sites.**