# Peer review of "Monitoring soil moisture from middle to high elevation in Switzerland: Set-up and first results from the SOMOMOUNT network"

_Hydrology and Earth System Sciences, 2016_

## Referee Comment (RC1) · Anonymous Referee #1 · 29 Sep 2016

This MS describes the new soil moisture monitoring network SOMOMOUNT launched in 2013 consisting of 6 soil moisture stations distributed along an altitudinal gradient between the Jura Mountains and the Swiss Alps.

Soil moisture monitoring in areas with low sensor density like Alpine regions is important e.g. for validation of global models and remote sensing products. Thus, it fits well to the scope of this journal. The MS is mostly written in an understandable way. However part of the text is not well comprehensible and too speculative. Also, there are major issues regarding the methods and interpretations of the results (see comments below).

General comments:

[Figure]

1) Soil water content measurements This study uses electromagnetic (EM) sensors to measure soil water content. It is very important to understand that this is an indirect measuring method and that EM sensors are only sensible to changes of the dielectric properties of the soil (i.e. the permittivity). To determine soil water content, EM sensors make use of the strong dependence of EM signal properties on volumetric water content that stems from the high permittivity of liquid water (∼80) compared to mineral solids (2–9), and air (1), see e.g. Bogena et al., 2007. However, it is well known that the permittivity of pure ice is extremely lower compared to liquid water (∼2-3) (e.g. Aragones et al., 2010). Therefore, during frozen soil conditions, the EM signal will decrease considerably, while the total soil water content stays the same. In addition, typically not all liquid water freezes at soil temperatures below 0°C, depending on the temperature, salinity, initial moisture, and soil texture (e.g. Zhang et al., 2003). Thus, the EM sensor determines the apparent permittivity of a mixture of liquid water, ice, mineral solids and air, with their respective permittivities. Consequently, the senor calibration determined for unfrozen soil is not valid any more (see e.g. Watanabe and Wake, 2009). From these theoretical considerations it becomes clear that the EM derived volumetric soil water content data shown in the EM during frozen soil conditions is not correct. This also means that the interpretations of the data are, at least partly, incorrect.

These problems are also very important with respect to publishing the data for validation purposes of global models. Clearly, a comparison of the erroneous volumetric soil water contents presented in this MS with model results will lead to deceptive deviations for frozen soil conditions.

Consequently, the authors either need to calculate the total soil water content, e.g. using the expanded dielectric mixing model presented by Watanabe and Wake (2009) or otherwise they would need to restrict their analysis to periods without soil freezing.

2) EM sensor calibration The authors used TDR sensors as reference for the SMT100 sensors. However, they only used the empirical, soil unspecific function of Topp et al.

(1980) to relate the permittivity measurements to soil water content. However, many studies showed that depending on the soil properties this can lead to uncertainties in the soil water content estimates (e.g. Robinson, 2004). Thus, the reference quality of the TDR data is questionable. The more advanced calibration approach presented by Rosenbaum et al. (2010) using the CRIM model would be more preferable to determine reference soil water content data.

3) Interpretation of the data The authors used so-called moisture orbits to analyse the soil water content data and to determine the dominate soil hydraulic processes. This is a quite appealing way to present the soil water content data and interesting pattern were shown. However, the interpretation of these patterns is at times very speculative and partly unrealistic if not completely wrong (see specific comments for examples).

Specific comments

P3 L20: This paper is not accessible. You should refer to the paper of Qu et al. (2013), which thoroughly described and tested the SPADE sensor, which is the successor of the SISOMOP sensor and precursor of the SMT100. All three sensor types are using exactly the same technique (ring oscillator) and only differ in their specific design (e.g. plastic material, sensor output etc.).

P4 L4-7: I do not think that the technical description is fully correct. See Evett et al. (2006) for a detailed description of the Delta-T PR1/6 Profiler (precursor of the PR2/6).

P8 L14: Arithmetic mean?

P9 L24-25: Different relationships might be due to specific soil properties. Since this is a purely empirical approach, I don't see the need for using the linear model for reasons of consistency. Instead the best model should be used to provide the best relationship between sensor output and soil water content.

P10 L6-7: This is not plausible. Deviations are more likely due to small scale heterogeneities of the soil. Please remember that the SMT100 is only sensible to EM

properties of the soil directly surrounding the sensor blade (just some mm to cm).

P10 L10-14: Again not plausible. The deviations are due to the different soil properties, which directly influence the measurements.

P10 L14 (K instead of °C)

P10 L14-17: Unclear which physical processes you are referring to.

P10 L28-29: Unclear why freezing should freezing and thawing explain the differences. The wet bias could be explained by the unreliable TDR calibration using the Topp equation.

P11 L12: Unreliable soil water content measurement due to frozen water.

P13 L20: What do you mean with "transfer"

P13 L25-27: Which sensors were used? How did you derive the soil water contents for the profiles (simple arithmetic mean, weighted mean etc.)?

P14 L1-4: This is an indication for preferential flow processes. Dry top soils tend to bypass precipitation water (see e.g. Wiekenkamp et al., 2016).

P14 L5-6: Difference in precipitation sums is too small to explain this difference. P14 L7-9: Not plausible. Evapotranspiration rates in this climate are mainly depending on meteorological forcing and vegetation characteristics.

P14 L13: This is counterintuitive. Why should melting start underground?

P14 L28-29: Not plausible (see above).

P14 L30-31: Not plausible. There seems rather to be a constant groundwater influence at 50 cm depth.

P15 L1-2: This statement is too general (in structured soil Ks is not always lower in greater depths due to preferential flow).

P15 L18: You need to mention that you are now showing the differentials.

P15 L25-26: Not plausible. In that case, the sensor in 50 cm would show an increase.

P15 L30: What is typical for this soil?

P16 L1-3: Not plausible. From basic soil physics is well known that the soil hydraulic conductivity decreases with decreasing soil water content. However, in structured soils preferential flow can be activated (see e.g. Wiekenkamp et al., 2016).

P16 L7-9: Not plausible why these orbit patterns should indicate lateral processes.

P17 L28-34: Not plausible why these processes can produce lower soil temperatures, although the air temperature is relative high. Are you measuring air temperature farther away from the soil station?

P18 L5-7: Not plausible (see above)

P18 L7-8: The soil seems not have been frozen during these phases (values are still relatively high)

Literature

Aragones J. L., L. G. MacDowell, and C. Vega (2010): Dielectric Constant of Ices and Water: A Lesson about Water Interactions. J. Phys. Chem. A 2011, 115, 5745–5758.

Bogena, H.R., J.A. Huisman, C. Oberdörster and H. Vereecken (2007): Evaluation of a low-cost soil water content sensor for wireless network applications. Journal of Hydrology, 344 (1-2): 32-42.

Evett S.R., J.A. Tolk, and T.A. Howell (2006): Soil Profile Water Content Determination: Sensor Accuracy, Axial Response, Calibration, Temperature Dependence, and Precision. Vadose Zone Journal 5:894–907.

Qu, W., H.R. Bogena, J.A. Huisman and Vereecken (2013): Calibration of a novel low-cost soil water content sensor based on a ring oscillator. Vadose Zone J. 12(2), doi:

[Figure]

10.2136/vzj2012.0139.

Robinson, D. A. (2004): Measurement of the solid dielectric permittivity of clay minerals and granular samples using a time domain reflectometry immersion method, Vadose Zone J., 3(2), 705–713, doi:10.2136/vzj2004.0705.

Watanabe, K. and Wake, T. (2009): Measurement of unfrozen water content and relative permittivity of frozen unsaturated soil using NMR and TDR. Cold Regions Science and Technology 59: 34–41

Wiekenkamp, I., J.A. Huisman, H. Bogena, H. Lin and H. Vereecken (2016): Spatial and Temporal Occurrence of Preferential Flow in a Forested Headwater Catchment. J. Hydrol. 534: 139-149, doi:10.1016/j.jhydrol.2015.12.050.

Zhang, L., J. Shi, Z. Zhang and K. Zhao (2003): The estimation of dielectric constant of frozen soil-water mixture at microwave bands. Geoscience and Remote, doi:10.1109/IGARSS.2003.1294626

---

## Referee Comment (RC2) · Anonymous Referee #2 · 10 Nov 2016

**Review of the Article hess-2016-474**

**Monitoring soil moisture from middle to high elevation in Switzerland: Set-up and first results from the SOMOMOUNT network**

**by C. Peller and C. Hauck**

This paper presents soil moisture observations collected along an altitudinal and climatic transect in Switzerland. The soil moisture dynamics is discussed (mostly qualitatively) with respect to elevation, soil properties and climate.

**General comments:**

I have mixed feelings about this paper.

On the one hand, it is a very nice paper to read. The soil moisture observations and the calibration procedures are well presented, in a didactic way. Observations are also interpreted in a clever way, the reasons of the differences among the stations clearly identified. At the end, the paper provides a good conceptual scheme to understand the role of elevation in controlling the yearly soil moisture cycle. I´m working with a similar soil moisture dataset collected in an Alpine region, and I recognize similar trends.

On the other hand, most of the results are based only on qualitative interpretations, and the paper does not add very new concepts on what is already known on the soil moisture behavior in cold/mountain climates.

I also share the doubts of the first reviewer on the unreliability of soil moisture observations when the soil is frozen. It is not so straightforward assuming that soil immediately freezes with negative soil temperatures. Soil could remain unfrozen in small pores also well below 0 C. Also assuming that TDR devices correctly detect low soil moisture when the soil is frozen it is also not so obvious. All the discussion is quite misleading on this point. You cannot consider the same situation having low liquid (but high total) water content due frozen soil and having a low total water content because the soil is dry. Those are completely different situations of low (liquid) water avalability, and this should be clearly differentiated in the discussion.

Moreover, reading the introduction the paper seems here really focused on permafrost. Also the broad literature review on mountain SWC dynamics is biased toward permafrost. Then, later in the results and discussion sections, the topic permafrost is barely covered.   I suggest a broader motivation. There are other important reasons to monitor mountain soil moisture, runoff production, climatic impacts, vegetation seasonality …

Nevertheless, I would recommend publication after a careful revision, since the paper could provide useful guidelines on how to interpret soil moisture observations in Alpine regions.

**Specific comments:**

**Abstract**

Please add more concrete results on which are the main soil moisture patterns.

**Introduction**

**Line 25-30.** The paper seems really focusing on permafrost. Then, later in the results section, the topic permafrost is hardly covered. I suggest a broader motivation. There are other important reasons to monitor mountain soil moisture, from runoff production to climatic impacts to vegetation seasonality.

**Instruments**

This section is too long, with a long revision of the advantages and disadvantages of different measurement techniques and many details on the calibration procedures. This part could become shorter. However, I found this part useful. I learned something new.

**Network design**

**Line 25-30.** This part could become shorter.

**Field sites**

**Pages 5-6-7** This part could also become shorter, moving more details to Table 3.

**Soil moisture temporal evolution.** This paragraph is very qualitative, but well written. However, I share here with Reviewer 1 some doubts on the methodology.
It is not so straightforward assuming that soil freezes when soil temperatures are negative. Soil could remain unfrozen in small pores also below 0. Also assuming that a TDR device measures low soil moisture when the soil is frozen it is not so obvious. See the comments of the other reviewer on this point.

**Page 11, Line 34.** "*During the snow melt period only a small VWC increase is seen, which could be attributed to conditions close to saturation throughout the winter.*" Interesting observation, but it would be nice to quantify better the impact of snow melt on initial VWC in spring. It is a relevant research question in the context of climate change.

**Page 12, Line 5.** Simply stating that VWC is minimum is not correct. The water is in the soil, but likely frozen. It is more correct to specify minimum liquid water content. Therefore, the whole the discussion is quite misleading. You cannot consider in the same way having low liquid (but high total) water content due frozen soil and having a low total water content because the soil is dry. Those are completely different ways to have low liquid water content.

**Soil moisture spatial distribution.** This paragraph is not very informative. It only informs us that there is (as expected) a large spatial variability, and where falls more rain is more wet. I suggest to skip or strongly reduce.

**Page 14, Line 28.** How thick is the organic layer? What about organic soil in the other sites? Soil type dependency is not discussed for all the sites. Why?

**Page 15, Line 4.** Do you mean Figure 10?

**Page 16, Line 3 and line 20** "*From Fig. 6c, it can be seen that, at the time of the precipitation event, the VWC at 10cm and 30cm depths were unusually low, enabling the water to pass through easily.*"
"*As for MLS above, the VWC is low at all depths enabling the precipitations to infiltrate quickly down to the deepest layers*"
Those statements apparently contradict hydrologic theory. In fact, larger is the water content, larger is the soil hydraulic conductivity and therefore larger is the amount of water that can infiltrate.
Please motivate your statements. Do you refer to the speed of the infiltration front or to the total amount of water infiltrated? (i.e. see Green – Ampt theory?). It is not only because if the soil is already wet you do not see a big change in soil moisture, but the infiltration amount is already significant?

**Page 16, Line 18.** Do you mean Figure 10?

**Altitude dependency.** This part of the paper is well written and informative, but very qualitative. This altitudinal dependency is already known in literature. Even if it is important to show how the new data published in this paper follow or not follow what is known in literature, a more robust quantitative analysis would be beneficial for the paper.

**Page 16, Line 28 .** "*Below 2000 m.a.s.l. the maximum VWC is recorded in winter and the minimum in summer, whereas above this threshold the inverse occurs (maximum VWC in summer and minimum in winter).*"
This refers only to the liquid water content. The total (frozen + unfrozen) remains high all the winter even at high elevations. Please specify this better.

**Page 17, Line 5-6 and following.** Ok, such processes are somehow clear, but … what are the implications for SWC?

**Page 17, Line 18** "*This is confirmed by the observed negative ground temperatures as well as the increasing freezing degree days. With increasing elevation, air and ground temperatures decrease yielding increasingly long duration of seasonally frozen ground and thus explaining the decreasing trend of VWC*"
Could you qualitatively estimate a functional relationship? Which is the temperature threshold? (I have seen that later the paper partially answers to those questions …)

**Page 17, lines 30 and followings.** Such "frost holes" are geological peculiarities well known in other places of the Alps (I.e. Kaltern, Italy; Lases, Italy). Their effect on SMC is interesting. May be there is more in the geological literature.

**Page 18, lines 25-32.** Well written, but it could be shortened. Too many unnecessary details.

**Conclusions** In general, I see here nothing on the implications on permafrost, which seems to be one of the major motivations of the paper. Either downplay this in the introduction, or develop more the permafrost topic in the discussion/conclusions sections.

**Page 19, lines 5-12.** This part should be moved to the method section. It partially addresses the major objection of Reviewer 1. You say "*Although the sensors are not specifically designed for freezing conditions, we found the measurements to be consistent, both regarding inter-sensor comparisons as well as in comparison with related variables such as ground temperature and precipitation*".
This should be better motivated in the method section.

**Page 19, lines 15-16.** I suggest to add here more details on the result that it does exist a SWC "peak" at about 2000 m a.s.l. This is an interesting result that should be highlighted.

**Figure 11.** I suggest to have the same x and y range in all the subplots.

---

## Author Comment (AC1) · 12 Dec 2016

*This MS describes the new soil moisture monitoring network SOMOMOUNT launched in 2013 consisting of 6 soil moisture stations distributed along an altitudinal gradient between the Jura Mountains and the Swiss Alps.*
*Soil moisture monitoring in areas with low sensor density like Alpine regions is important e.g. for validation of global models and remote sensing products. Thus, it fits well to the scope of this journal. The MS is mostly written in an understandable way. However part of the text is not well comprehensible and too speculative. Also, there are major issues regarding the methods and interpretations of the results (see comments below).*

**General comments:**
*Soil water content measurements: This study uses electromagnetic (EM) sensors to measure soil water content. It is very important to understand that this is an indirect measuring method and that EM sensors are only sensible to changes of the dielectric properties of the soil (i.e. the permittivity). To determine soil water content, EM sensors make use of the strong dependence of EM signal properties on volumetric water content that stems from the high permittivity of liquid water (_80) compared to mineral solids (2–9), and air (1), see e.g. Bogena et al., 2007. However, it is well known that the permittivity of pure ice is extremely lower compared to liquid water (_2-3) (e.g. Aragones et al., 2010). Therefore, during frozen soil conditions, the EM signal will decrease considerably, while the total soil water content stays the same. In addition, typically not all liquid water freezes at soil temperatures below 0_C, depending on the temperature, salinity, initial moisture, and soil texture (e.g. Zhang et al., 2003). Thus, the EM sensor determines the apparent permittivity of a mixture of liquid water, ice, mineral solids and air, with their respective permittivities. Consequently, the senor calibration determined for unfrozen soil is not valid any more (see e.g. Watanabe and Wake, 2009). From these theoretical considerations it becomes clear that the EM derived volumetric soil water content data shown in the EM during frozen soil conditions is not correct. This also means that the interpretations of the data are, at least partly, incorrect.*

*These problems are also very important with respect to publishing the data for validation purposes of global models. Clearly, a comparison of the erroneous volumetric soil water contents presented in this MS with model results will lead to deceptive deviations for frozen soil conditions.*

*Consequently, the authors either need to calculate the total soil water content, e.g. using the expanded dielectric mixing model presented by Watanabe and Wake (2009) or otherwise they would need to restrict their analysis to periods without soil freezing.*

We are thankful to the reviewer for this important and detailed comment. While we are fully aware of the limitations of the electromagnetic methods in case of frozen conditions, we have failed to explain it accordingly in our manuscript. In this paper, we only consider the liquid water content given that both the TDR and the FDR technique are unable to measure the total water content in frozen conditions unless a dielectric mixing model is applied. At the moment we do not have all the parameters needed for the application of such models and we feel that it would constitute a study in itself.

We agree with the reviewer that liquid VWC measured at temperature below the freezing point should be handled with caution and identified as such. The calibration procedure and resulting sensor accuracy described in this manuscript is only valid for ground temperature above 0°C. Under frozen conditions, the accuracy of absolute measurement is difficult to assess.  According to Watanabe and Wake (2009), for sands the application of Topp's empirical relationship in frozen

conditions shows only small deviations from the measured total VWC using NMR except for temperatures between 0 and -1°C.

However, from previous knowledge based on the continuous monitoring of ground electrical resistivity (based on Electrical Resistivity Tomography measurements) we are confident that the relative changes in liquid VWC are well captured. At Schilthorn, Hilbich et al. (2011) identified the same VWC (measured with similar FDR devices as in the present study) and temperature stages than the ones presented in this manuscript. These stages were compared to the measured apparent resistivity in the uppermost 50cm, which similarly to the electromagnetic technique is highly sensitive on the amount of unfrozen water content and is able to detect variations even at T <0°C (cf. Hauck 2002). This comparison showed that the timing of VWC increase measured with FDR sensors is consistent with resistivity decrease and the same stages (frozen, zero-curtain and unfrozen) were identified.

In order to fully answer this important comment in the manuscript we added an additional sub-section (3.1 *Technical considerations for frozen conditions*) after the calibration part:

*"3.1 Technical considerations for frozen conditions*

*Given the high-elevation application of the FDR and TDR techniques at field sites which undergo freezing and thawing processes, some considerations are important to make. As mentioned above both techniques make use of the high permittivity of liquid water (~80) compared to the surrounding soil and air (2-9 and 1, respectively) to relate the recorded electromagnetic signal to the VWC. However, under frozen conditions a part of this total water content turns into ice, which has a much lower permittivity (~2-3, e.g. Aragones et al., 2010). Thus, upon freezing, the recorded signal and measured VWC strongly decreases although the total VWC stays constant. Given these limitations, the term VWC used hereafter is always referring to the liquid VWC.*

*Characteristically, at temperatures below 0°C water and ice can coexist in the soil (e.g. Spaans and Baker, 1995). However, the calibration procedure presented was conducted at room temperature and thus does not account for the presence of ice in the soil mixture. The resulting sensor accuracy is therefore only valid for above 0°C ground temperature (unfrozen conditions). The use of standard empirical calibration in frozen conditions often yields overestimations of the liquid VWC (e.g. Spaans and Baker, 1995; Yoshikawa and Overduin, 2005). However, according to Watanabe and Wake (2009), for sand the calibration using Topp's empirical relationship in frozen conditions shows only small deviations from the measured total VWC using NMR except for temperatures between 0 and -1°C.*

*Although the absolute accuracy of measured liquid VWC under frozen conditions is difficult to assess, the relative changes are well captured. At SCH, Hilbich et al. (2011) showed that the soil apparent resistivity (using data from continuous ERT monitoring) and soil moisture (measured with similar FDR devices as in the present study) exhibit consistent variations under frozen and unfrozen conditions. Given the sandy composition of the ground at SCH and STO as well as the evidence from the coinciding resistivity measurements (cf. also Hauck 2002), we find the liquid VWC data to be consistent enough to be used here with the standard calibration described above. However, the VWC measurements carried out at temperatures below 0°C are clearly identified in all figures in the manuscript and have to be interpreted with care, especially regarding their absolute values."*

Additionally, using the standard temperature quality flag (Dorigo et al., 2013), we clearly identified all VWC measurement made at temperature below 0°C in all the relevant figures of the revised version (new versions of Figures 5-9). We also clearly stated that whenever the term VWC is used in the manuscript it always refers to liquid VWC. The manuscript was entirely reviewed to clarify, wherever necessary, that only liquid water content is considered.

Finally, instead of showing the detailed evolution of the liquid VWC at Stockhorn, we show the liquid VWC, snow and temperature evolution at Schilthorn including the measured specific resistivities of the uppermost 1m of the ground in the revised version of figure 7(see below). Similarly to the observation of Hilbich et al. (2011), the relative changes of VWC are matching the evolution of the electrical resistivities. A corresponding paragraph was included in the text:

*"Although, the accuracy of the soil moisture measurements during the frozen and zero-curtain periods is difficult to assess due to the presence of ice, the relative changes and thus the timing of each phase is well captured. The liquid VWC variations are coherent with the change of specific resistivity in the uppermost ~50 centimetres of the ground. Similar to dielectric permittivity, electrical resistivity is highly dependent on the amount of unfrozen water content in the ground (Hauck 2002). The frozen stage is thus characterized by high resistivities and a marked drop is observed during the zero-curtain period due to the thawing of the ground (see also Hilbich et al. 2011). Finally, the unfrozen stage exhibits the lowest resistivity values."*

[Figure]

**Fig. 7: Measured liquid VWC, ground temperature and soil resistivity at FRE (a) and SCH (b) from March to August 2015. In addition, daily air temperature, snow depth and precipitation sums are shown as well as the date of the transition between the different stages in the thermal evolution at SCH at 10cm (dashed lines A and B, see text for details). The dashed VWC lines represent the soil moisture measurements taken at ground temperature below 0°C.**

Hauck, C.: Frozen ground monitoring using DC resistivity tomography. Geophysical Research Letters, 29 (21): 2016, doi: 10.1029/2002GL014995, 2002.
Hilbich, C., Fuss, C., Hauck, C.: Automated time-lapse ERT for improved process analysis and monitoring of frozen ground, Permafrost and Periglacial Processes 22(4), 306-319, DOI: 10.1002/ppp.732, 2011.

*EM sensor calibration: The authors used TDR sensors as reference for the SMT100 sensors. However, they only used the empirical, soil unspecific function of Topp et al. (1980) to relate the permittivity measurements to soil water content. However, many studies showed that depending on the soil properties this can lead to uncertainties in the soil water content estimates (e.g. Robinson, 2004). Thus, the reference quality of the TDR data is questionable. The more advanced calibration approach presented by Rosenbaum et al. (2010) using the CRIM model would be more preferable to determine reference soil water content data.*

We thank the reviewer for this pertinent comment. We are aware of the limitation of the soil unspecific calibration function of Topp et al. (1980). However, in this study the TDR sensors are only used for inter-network comparison with the SwissSMEX network but not for further analysis. Thus, for consistency we adopted the same approach as Mittelbach et al. (2011) and used the built in calibration of the TDR sensors. The text regarding the use of TDR as reference was clarified accordingly. As for the FDR sensors, the VWC measurement carried out at temperatures below 0°C are clearly identified in all the figures (5 and 6).

**Original manuscript**
*"For the PICO64, the built-in calibration based on Topp's equation (Topp et al., 1980) was used and no additional material specific calibration was performed according to Mittelbach et al. (2011). In our study, the PICO64 is mainly used as reference sensor regarding future comparison of data between the SwissSMEX soil moisture network and the SOMOMOUNT network."*

**Revised version:**
*"For the PICO64, the built-in calibration based on Topp's equation (Topp et al., 1980) was used and no additional material specific calibration was performed. In this study the PICO64 sensors are only used for inter-network comparison with the SwissSMEX network but not for further analysis. Thus for consistency we adopted the same calibration approach as Mittelbach et al. (2011) (i.e. the built in calibration for generic soils)."*

*Interpretation of the data: The authors used so-called moisture orbits to analyse the soil water content data and to determine the dominate soil hydraulic processes. This is a quite appealing way to present the soil water content data and interesting pattern were shown. However, the interpretation of these patterns is at times very speculative and partly unrealistic if not completely wrong (see specific comments for examples).*

We are thankful to the reviewer to point out the critical passages and for the input to our discussion of the results. We carefully reviewed our interpretations of the different moisture orbits and revised the manuscript according to the specific comments (see answers below).

**Specific comments**

*P3 L20: This paper is not accessible. You should refer to the paper of Qu et al. (2013), which thoroughly described and tested the SPADE sensor, which is the successor of the SISOMOP sensor and precursor of the SMT100. All three sensor types are using exactly the same technique (ring oscillator) and only differ in their specific design (e.g. plastic material, sensor output etc.).*

We are thankful to the reviewer for pointing out this fact and modified the reference accordingly.

*P4 L4-7: I do not think that the technical description is fully correct. See Evett et al. (2006) for a detailed description of the Delta-T PR1/6 Profiler (precursor of the PR2/6).*

We thank the reviewer for this comment and modified the technical description of the PR2/6 according to Verhoef et al. (2006). It now reads:

*"Finally, the PR2/6 sensor is a 100cm long down-hole water content sensor measuring soil moisture at 6 different depths (10, 20, 30, 40, 60 and 100cm) using the capacitance technique (Fig. 2). Each measurement depth comprises a pair of stainless steel rings, which transmit the 100 MHz electromagnetic signal into the ground, and one detector, which records the returned signal. This technique relies on the fact that the emitted wave generates an electromagnetic field, which extends about 100mm into the surrounding soil and depending on its dielectric properties and thus on the VWC is partly reflected (Verhoef et al., 2006)."*

*P8 L14: Arithmetic mean?*

We are thankful to the reviewer for this question and clarified this point in the text.

*P9 L24-25: Different relationships might be due to specific soil properties. Since this is a purely empirical approach, I don't see the need for using the linear model for reasons of consistency. Instead the best model should be used to provide the best relationship between sensor output and soil water content.*

We thank the reviewer for this comment. We agree with the reasoning and changed the sensor calibration accordingly. The sensor outputs were calibrated at DRE using the exponential fit and all relevant figures and text parts have been updated with the new calibrated values.

*"Thus, the linear calibration is used for all sites since except at DRE, where the exponential one is preferred"*

*P10 L6-7: This is not plausible. Deviations are more likely due to small scale heterogeneities of the soil. Please remember that the SMT100 is only sensible to EM properties of the soil directly surrounding the sensor blade (just some mm to cm).*

We thank the reviewer for this comment and clarified the sentence accordingly.

**Original manuscript:**
*"Deviations from the one-to-one correspondence of the two sensors (black line) can be attributed to time delays in infiltration events and/or evaporation events."*

**Revised version:**
*"Deviations from the one-to-one correspondence of the two sensors (black line) can be attributed to small scale heterogeneities in the soil directly surrounding the sensors, which can result in differences of reaction time to the infiltration and/or evaporation events as well as wet or dry bias due to different soil properties"*

*P10 L10-14: Again not plausible. The deviations are due to the different soil properties, which directly influence the measurements.*

We are thankful to the reviewer for pointing out this fact. The mentioned section was modified accordingly.

**Original manuscript:**
*"At DRE (Fig. 4b) the right sensor shows consistently higher values (~5-10 vol.%) than the left sensor, but the relation between the measured VWC values is almost constant. This is due to the soil composition (sandy loam rich in organic matter and with a very low bulk density), which has a high*

*hydraulic conductivity leading to almost instantaneous increase in VWC following precipitation events.”*

**Revised version:**
*“At DRE (Fig. 4b) the right sensor shows consistently higher values (~5-10 vol.%) than the left sensor, but the relation between the measured VWC values is almost constant, illustrating the effect of different soil properties.”*

*P10 L14 (K instead of °C)*

We modified the manuscript accordingly.

*P10 L14-17: Unclear which physical processes you are referring to.*

The text was modified by explicitly adding the physical processes we are referring to.

*“DRE is also the site with the largest RMSE (0.566°K) between the measured temperatures at 30cm depth, indicating that specific physical processes **such as the convective heat transport through air flow** (e.g. Wicky and Hauck 2016) may influence the two sensors in a different way (see Sect. 5.3).”*

Wicky, J., & Hauck, C.: Numerical modelling of convective heat transport by air flow in permafrost-affected talus slopes. The Cryosphere Discuss., 1–29, doi:10.5194/tc-2016-227, 2016.

*P10 L28-29: Unclear why freezing and thawing should explain the differences. The wet bias could be explained by the unreliable TDR calibration using the Topp equation.*

We thank the reviewer for this helpful comment. As seen in Fig.6 and mentioned in our above answer to the major comments, the largest changes in VWC at SCH and STO occur during the freezing/thawing of the ground. The observed relative VWC changes have been found to be consistent with ERT measurements and thus we believe them to be well captured. Due to soil heterogeneities and differential influence of the snow cover, the onset of freezing/thawing can differ significantly even at very small spatial scale, which creates easily strong deviations in determined unfrozen water content values at the sensor positions (e.g. Scherler et al. 2010, Hilbich et al. 2011, Pellet et al. 2016). The identification of the frozen and unfrozen period in the figure shows that the timing of the major deviations corresponds to the thawing period of the ground, further supporting our interpretation.

However concerning the wet bias at SCH we share the point of view of the reviewer and changed the text accordingly.

**Original manuscript:**
*“At SCH and STO the differences between the sensors have a characteristic shape, but are centred on the one-to-one relation. It can be attributed to different onset of freezing and thawing processes at the two sensor locations (see also Fig. 6e-f). Additionally, a clear wet bias in the PICO64 measurements can be observed at SCH.”*

**Revised version:**
*“At SCH and STO the differences between the sensors have a characteristic shape, but are centred on the one-to-one relation. It can be attributed to different onset of freezing and thawing processes at the two sensor locations marked by the grey dots (see also Fig. 6e-f). Additionally, clear wet and dry biases in the PICO64 measurements are observed at SCH and STO, respectively, which can be*

*explained by an unreliable calibration using Topp's equation for the high-mountain subsurface material present at SCH and STO (e.g. Robinson et al., 2004)."*

*P11 L12: Unreliable soil water content measurement due to frozen water.*

The absolute values may be unreliable during freezing (see our answer to **general comments**), but the period of minimum values in winter is robust and not uncertain, as show the complementary data sets from electrical resistivity (inverse of conductivity) measurements. These were now included in the revised version of Figure 7.

*P13 L20: What do you mean with "transfer"*

The manuscript was modified in order to clarify this sentence.

**Original manuscript:**
*"The shape of the resulting point cloud depends on the nature and speed of the water transfer processes, as well as on soil properties such as hydraulic conductivity, degree of saturation and porosity."*

**Revised version:**
*"The shape of the resulting point cloud depends on the nature and speed of the vertical transport of water through the soil layers, as well as on soil properties such as hydraulic conductivity, degree of saturation and porosity."*

*P13 L25-27: Which sensors were used? How did you derive the soil water contents for the profiles (simple arithmetic mean, weighted mean etc.)?*

We thank the reviewer for this comment and specified in the text that only the SMT100 sensors have been used for the all moisture orbit analysis. The profile probe and the PICO64 were not considered.

**Original manuscript:**
*"To investigate the seasonal dynamic of soil moisture, we used the moisture orbits between 10 and 50 cm depth for the year 2015 (Fig. 9)."*

**Revised version:**
*"To investigate the seasonal dynamic of soil moisture, we used the moisture orbits between the SMT100 sensors installed at 10 and 50 cm depth for the year 2015 at each site (Fig. 8)."*

*P14 L1-4: This is an indication for preferential flow processes. Dry top soils tend to bypass precipitation water (see e.g. Wiekenkamp et al., 2016).*

We are thankful to the reviewer for this comment and changed the text accordingly.

**Original manuscript:**
*"It indicates that at MLS the near surface VWC did not recover from the increased evaporation generated by the heat wave in July 2015 (cf. Fig. 6). "*

**Revised version:**
*"It indicates that at MLS the near surface VWC did not recover from the increased evaporation generated by the heat wave in July 2015 (cf. Fig. 6) creating suitable conditions for preferential flow processes due to the dry top soil, which tends to induce bypass flows of precipitation water (e.g. Wiekenkamp et al., 2016)."*

*P14 L5-6: Difference in precipitation sums is too small to explain this difference.*
*P14 L7-9: Not plausible. Evapotranspiration rates in this climate are mainly depending on meteorological forcing and vegetation characteristics.*

We agree with the reviewer and carefully revised our interpretation of the effects of the July 2015 heat wave observed at FRE and MLS. Looking at the measured air temperature and incoming radiation, one can see that the meteorological anomaly during heat wave was of similar amplitude at both sites. Furthermore, according to MeteoSwiss the calculated potential evaporation for the three weeks period is in the same range (186mm at MLS and 203mm at FRE). Thus the larger VWC decrease observed at MLS can be attributed to a lower limitation of actual evaporation than at FRE due to higher initial VWC. The manuscript was modified as follow:

**Original manuscript:**
*"On the contrary, at FRE the VWC returned to its original value. This can be explained by two main factors namely the precipitation regimes and the soil properties. MLS received less precipitation than FRE (1036 mm $y^{-1}$ and 1254 mm $y^{-1}$ respectively in 2015, MeteoSwiss), which hampered the full recovery of soil moisture conditions after the heat wave. Additionally, the absolute quantity of water lost at MLS was much larger due to stronger evaporation. The amount of water available for evaporation is dependent on the soil properties. At MLS the soil type is silty loam and is able to retain more water than the sandy loam (soil type at FRE)."*

**Revised version:**
*"On the contrary, at FRE the VWC returned to its original value. Comparatively, both sites received a similar amount of precipitation following the heat wave, from July to end of 2015 (443 mm and 534 mm respectively, MeteoSwiss). Furthermore, the atmospheric forcing (i.e. air temperature, radiation and calculated potential evaporation) was very similar at both sites (MeteoSwiss). Therefore, the larger and longer lasting impact of the 2015 heat wave at MLS is due to the higher initial VWC and thus lower potential evaporation limitation than at FRE. The amount of water available for evaporation is dependent on the soil properties. At MLS the soil type is silty loam and is able to retain more water than the sandy loam present at FRE.*

*P14 L13: This is counterintuitive. Why should melting start underground?*

There are two processes that can cause the ground to start melting at larger depth: the preferential infiltration and refreezing of water at certain depths (e.g. by slope processes) or the influence of the heat penetration from the preceding summer. The heat from the summer penetrates in the ground and propagates by heat conduction to larger depths. Due to the time lag of this propagation to larger depths and depending on the strength of the winter freezing, the temperatures at depth can potentially be warmer than near the surface, and thus melting will start from below. This process is especially marked for permafrost temperature close to 0°C as it is the case at SCH. A description of this process can be found e.g. in Zenklusen Mutter and Phillips (2012). The manuscript was modified as follows:

**Original manuscript:**
*"This is followed by a sharp increase at 50cm not seen at 10cm (1), followed by a strong increase at 10cm but not at 50cm (2) consistent with the melting of the ground from underneath, which takes place at different time at the two depths (spring zero-curtain)."*

**Revised version:**
*"This is followed by a sharp increase at 50cm not seen at 10cm (1), followed by a strong increase at 10cm but not at 50cm (2) consistent with the melting of the ground from underneath, which takes*

*place at different time at the two depths (spring zero-curtain). The thawing of the ground from below can be due to preferential water infiltration events or to the influence of warm temperatures from the preceding summer at depth (e.g. Zenklusen Mutter and Phillips, 2012). In the latter, due to the time-lag of the heat propagation at depth, warmer ground temperatures can be found at depth, thus starting the melting from below. This process is especially marked for permafrost temperature close to 0°C as it is the case at SCH."*

*P14 L28-29: Not plausible (see above).*

The manuscript was adapted as follow.

**Original manuscript:**
*"Furthermore, and in contrast to the soil type below, the organic rich material at the surface retains the water and lead to enhanced summer evaporation and winter freezing."*

**Revised version:**
*"Furthermore, and in contrast to the large blocks below, the organic rich material at the surface retains the water and has a large thermal conductivity (Beringer et al., 2001), thus favouring the summer evaporation and winter freezing.*"

*P14 L30-31: Not plausible. There seems rather to be a constant groundwater influence at 50 cm depth.*

After careful revision of the ancillary measurements at our disposal, we agree with the interpretation of reviewer1. The manuscript was adapted as follow:

**Original manuscript:**
*"At GFU the presence of an organic rich layer in the uppermost 10cm of the soil causes the measured VWC at 10cm to be highly variable and higher than in the remaining soil column yielding an almost horizontal moisture orbit shape. "*

**Revised version:**
*"At GFU the presence of an organic rich layer in the uppermost 10cm of the soil causes the measured VWC at 10cm to be highly variable and higher than in the remaining soil column (see also Fig. 6d). At 50cm the measured VWC show near saturation conditions throughout the year indicating a potential influence of shallow ground water. This is confirmed by additional ERT measurements realized in summer from 2013 to 2015, which indicate extremely low specific resistivities (~250 Ωm) down to 2.5m (not shown). The combination of near saturated conditions at 50cm and highly variable VWC at 10cm yields an almost horizontal moisture orbit shape. "*

*P15 L1-2: This statement is too general (in structured soil Ks is not always lower in greater depths due to preferential flow).*

This statement was relativized as follows:

**Original manuscript:**
*"Furthermore the soil types at 30cm and 50cm have a lower hydraulic conductivity (Cosby et al., 1984), which also contributes to the lower soil moisture variability at these depths"*

**Revised version:**
*"Furthermore, at 30 and 50cm the soil is composed of loam and sandy loam with much larger bulk densities (Table 3), which are typically characterized by lower hydraulic conductivities if no*

*preferential flow is occurring (Cosby et al., 1984). These soil properties also contribute to the observed lower soil moisture variability at these depths"*

*P15 L18: You need to mention that you are now showing the differentials.*

We thank the reviewer for this remark and modified the text accordingly.

*P15 L25-26: Not plausible. In that case, the sensor in 50 cm would show an increase.*

We thank the reviewer for this comment and revised our interpretation as follow.

**Original manuscript:**
*"At SCH the maximum VWC at 10cm is reached after two hours while no variation is recorded at 50cm. This pattern is typical for highly draining soils such as sand (found at SCH). "*

**Revised version:**
*"At SCH the maximum VWC at 10cm is reached after two hours while no variation is recorded at 50cm during that interval. The VWC starts increasing at 50cm once the VWC at 10cm is already decreasing. This pattern is consistent with the vertical succession of soil found at SCH: sandy loam at the surface, which retains water at the beginning of the event and sand at larger depth, which is more draining."*

*P15 L30: What is typical for this soil?*

At DRE one has to keep in mind that the particular soil moisture behaviour is not really typical for one soil type but it is particular to the whole profile (including the coarse blocky talus slope below). The sentence was clarified as follow.

**Original manuscript:**
*"It indicates a rapid transfer of water through the soil and no storage at 10cm typical for the particular soil composition found at DRE."*

**Revised version:**
*"It indicates a rapid transfer of water through the soil and little storage at 10cm, which is typical for the particular soil composition found at DRE (single organic rich layer with low bulk density underlain by coarse blocks with large interconnected pores). "*

*P16 L1-3: Not plausible. From basic soil physics is well known that the soil hydraulic conductivity decreases with decreasing soil water content. However, in structured soils preferential flow can be activated (see e.g. Wiekenkamp et al., 2016).*

We thank the reviewer for this comment and changed the sentence accordingly.

**Original manuscript:**
*"From Fig. 6c, it can be seen that, at the time of the precipitation event, the VWC at 10cm and 30cm depths were unusually low, enabling the water to pass through easily."*

**Revised version:**
*"From Fig. 6c, it can be seen that, at the time of the precipitation event, the VWC at 10cm and 30cm depths were unusually low, thus providing very suitable conditions for the activation of preferential flow (see e.g. Wiekenkamp et al., 2016)."*

*P16 L7-9: Not plausible why these orbit patterns should indicate lateral processes.*

Fig. 11e does not necessarily indicate lateral processes, but show the influence of snow melt processes. As water infiltration from snow melt is a spatially heterogeneous process on slopes, its potential influence is mentioned here. The paragraph was reformulated accordingly.

**Original manuscript:**
*"An example of these lateral processes can be seen at STO, where the precipitation event shown in Fig. 11e did not yield a clear moisture orbit."*

**Revised version:**
*"The infiltration of snow melt water is a spatially very heterogeneous process on slopes, especially when the subsurface is characterized by large size particles and draining soil types. An example of the influence of snow melt processes can be seen at STO, where the precipitation event shown in Fig. 10e did not yield a clear moisture orbit."*

*P17 L28-34: Not plausible why these processes can produce lower soil temperatures, although the air temperature is relative high. Are you measuring air temperature farther away from the soil station?*

The colder ground surface temperature at the bottom of the slope is due to the convective heat transport by air flow through the available pore space within the talus slope.  In winter, the lower air temperature outside compared to inside the talus provokes the apparition of a pressure gradient and thus leads to ascending warm air in the talus compensated by cold air inflow at the bottom of the slope. During the summer (i.e. when the temperature of the air inside the talus is lower than outside), the circulation is reversed and gravitational cold air outflow is taking place at the bottom of the slope. This typical seasonally reversing air circulation was first described by Wakonigg (1996) and identified at DRE by Delaloye (2004) and Morard (2011). It was further confirmed through model studies (e.g. Wicky and Hauck 2016).

The paragraph relative to the cooling effect of the air circulation was reformulated as follows in order to clarify the process and to detail the location of the soil moisture and weather station.

**Original manuscript:**
*"The low mean annual ground temperature results from a complex air circulation within the underlying talus slope (Delaloye, 2004; Morard, 2011), which is made possible by the large interconnected pore space between the coarse blocks of the talus. During winter, ascending warm air within the talus slope leads to cold air inflow at the bottom of the talus slope, where the soil moisture station is located. This process efficiently cools the ground even when the snow cover is present and has been observed at many similar talus slopes in low and high mountain regions (e.g. Delaloye and Lambiel 2005; Gude et al. 2003; Kneisel et al. 2000; Sawada et al. 2003; Wakonigg, 1996). In summer the reverse process takes place with outflow of cold air from the inside of the talus slope at the bottom by gravity. "*

**Revised version:**
*"The low mean annual ground temperature results from convective heat transport by a complex air circulation within the underlying talus slope (Delaloye, 2004; Morard, 2011), which is made possible by the large interconnected pore space between the coarse blocks of the talus. During winter, ascending warm air within the talus slope leads to cold air inflow at the bottom of the talus slope, where the soil moisture and weather stations are located. This process is able to efficiently cool the ground even when the snow cover is present. In summer the air circulation is reversed and a gravity-driven outflow of cold air from the inside of the talus slope takes place at the bottom, where soil moisture is measured. This process has been observed at many similar talus slopes in low and high*

*elevation mountain regions (e.g. Delaloye and Lambiel 2005; Gude et al. 2003; Kneisel et al. 2000; Sawada et al. 2003; Wakonigg, 1996). Furthermore, the lower ground temperatures caused by the air circulation have been successfully reproduced using numerical modelling (Wicky and Hauck, 2016).*"

*P18 L5-7: Not plausible (see above)*

See the answer to the previous comment

*P18 L7-8: The soil seems not have been frozen during these phases (values are still relatively high)*

During the winter 2014-2015 below 0°C temperatures have been measured at 10cm. At 30cm and 50cm as well as during the winter 2015-2016 the measured temperatures are slightly above 0°C at all depths but these values are still within the sensor accuracy (±0.2°C). Furthermore the observed liquid VWC decrease in winter is consistent with at least partial freezing of the ground. The text was rephrased accordingly.

**Original manuscript:**
*"Thus two phases of minimal VWC values are observed in summer and winter (see Fig. 6b)."*

**Revised version:**
 *"Thus two phases of minimal VWC values are observed: one during the summer due to evaporation and one in winter due to partial or complete freezing of the ground (see Fig. 6b). "*

**Revised figures and captions:**

[Figure]

**Fig. 5: Comparison between TDR- (x-axis) and FDR-measured liquid VWC (y-axis) at all sites. The linear relation is used for the FDR calibration. The hollow grey points at GFU, SCH and STO represent soil moisture measurements taken when the ground temperature was below 0°C.**

[Figure]

**Fig. 6: Measured VWC (upper panel) and ground temperatures (lower panel) at each SOMOMOUNT station (a-f). At FRE the uppermost panel displays the PR2/6 measured liquid VWC. The vertical dotted lines at FRE and STO indicate the period analysed in Fig. 7. The dashed VWC lines represent the soil moisture measurements taken when the ground temperature was below 0°C.**

[Figure]

**Fig. 7: Measured liquid VWC, ground temperature and soil resistivity at FRE (a) and SCH (b) from March to August 2015. In addition, daily air temperature, snow depth and precipitation sums are shown as well as the date of the transition between the different stages in the thermal evolution at SCH at 10cm (dashed lines A and B, see text for details). The dashed VWC lines represent the soil moisture measurements taken at ground temperature below 0°C.**

[Figure]

**Fig. 8: Moisture orbit at each SOMOMOUNT station from the 1st January to the 31st December 2015. The numbered arrows indicate the most important stages at each station as well as the sense of the evolution. The hollow circles represent soil moisture measurements taken when the temperature was below 0°C at 50cm.**

[Figure]

**Fig. 9: Moisture orbits at FRE (a), GFU (b) and SCH (c) for the consecutive monitoring years (Jan. 2014-Aug. 2016 at FRE, Aug. 2013-Aug. 2016 at GFU and Aug. 2014-June 2016 at SCH). The hollow circles represent soil moisture measurements taken when the temperature was below 0°C at 50cm.**

[Figure]

Fig. 12: Elevation dependency of the winter-, summer- and annual mean liquid VWC at 30cm depth (a), air-, ground temperature and surface offset (ground minus air temperature) at 10cm (b), annual precipitation sum including selected SwissMetNet stations for comparison (cf. Fig. 1) (c), freezing and thawing degree days (calculated from ground temperatures at 10cm) (d) and snow duration (calculated from ground temperature at 10cm using the method described in Staub and Delaloye (2016)) (e) during the year 2015. The VWC values for SIO, PAY and PLA are part of the SwissSMEX network (Mittelbach and Seneviratne, 2012). Due to missing data at 30cm the VWC shown for PAY and PLA was measured at 50cm and 20cm at CER. The dashed green lines illustrate the linear regression based on all available SwissMetNet, PERMOS and IMIS stations in Switzerland (the numbers of station with complete data series in 2015 are indicated) and the shaded areas represent the 99% confidence intervals. The length of each regression line corresponds to the maximum elevation of the available stations.

**Review of the Article hess-2016-474**
**Monitoring soil moisture from middle to high elevation in Switzerland: Set-up and first results from the SOMOMOUNT network**
**by C. Pellet and C. Hauck**
*This paper presents soil moisture observations collected along an altitudinal and climatic transect in Switzerland. The soil moisture dynamics is discussed (mostly qualitatively) with respect to elevation, soil properties and climate.*

**General comments:**
*I have mixed feelings about this paper.*
*On the one hand, it is a very nice paper to read. The soil moisture observations and the calibration procedures are well presented, in a didactic way. Observations are also interpreted in a clever way, the reasons of the differences among the stations clearly identified. At the end, the paper provides a good conceptual scheme to understand the role of elevation in controlling the yearly soil moisture cycle. I´m working with a similar soil moisture dataset collected in an Alpine region, and I recognize similar trends.*
*On the other hand, most of the results are based only on qualitative interpretations, and the paper does not add very new concepts on what is already known on the soil moisture behavior in cold/mountain climates.*
*I also share the doubts of the first reviewer on the unreliability of soil moisture observations when the soil is frozen. It is not so straightforward assuming that soil immediately freezes with negative soil temperatures. Soil could remain unfrozen in small pores also well below 0 C. Also assuming that TDR devices correctly detect low soil moisture when the soil is frozen it is also not so obvious. All the discussion is quite misleading on this point. You cannot consider the same situation having low liquid (but high total) water content due frozen soil and having a low total water content because the soil is dry. Those are completely different situations of low (liquid) water availability, and this should be clearly differentiated in the discussion.*

We thank the reviewer for pointing out these facts. We fully agree that the discussion in the submitted manuscript was quite misleading concerning the distinction between total and liquid water content. As mentioned in our answer to reviewer 1's comment we carefully revised the manuscript and clearly stated that whenever we use the term VWC, liquid VWC is meant given the methods limitations.

We also thank both reviewers for sharing their concerns regarding the methods used here and their application in frozen conditions. On that topic a detailed answer was made to reviewer 1 and a whole new subsection was included after the calibration part. Similarly, all figures with VWC data during frozen conditions were revised in order to illustrate more clearly, where freezing occurred and uncertainties regarding the VWC calibration exist due to freezing. Furthermore, additional resistivity data have been included in figure 7 to illustrate our statement that the relative liquid VWC changes are well captured although the absolute values of liquid VWC is difficult to assess in frozen conditions.

*Moreover, reading the introduction the paper seems here really focused on permafrost. Also the broad literature review on mountain SWC dynamics is biased toward permafrost. Then, later in the results and discussion sections, the topic permafrost is barely covered. I suggest a broader motivation. There are other important reasons to monitor mountain soil moisture, runoff production, climatic impacts, vegetation seasonality …*

We are thankful to the reviewer for this comment. We modified the introduction and broadened the motivations for behind this study and downplayed the permafrost part. The text reads now as follows:

*"In mountain environments, soil moisture is particularly crucial since it can control the initiation of convective precipitation (e.g. Barthlott et al., 2011; Hauck et al., 2011), the generation of runoff (e.g. Morbidelli et al., 2016; Zehe et al., 2010) and thereby the mitigation or intensification of flash floods (e.g. Borga et al., 2007). Soil moisture also significantly affects the vegetation growth and distribution (e.g. Paschalis et al., 2015; Porporato et al., 2004). In terrains affected by seasonal and permanently frozen conditions, its effect on the stability of slopes and the thermal and kinematic characteristics of periglacial landforms was highlighted in several observation and modelling studies (e.g. Boike et al., 2008; Hasler et al. 2011; Hinkel et al., 2001; Krautblatter et al. 2012; Scherler et al., 2010; Streletskiy et al. 2014; Westermann et al., 2009; Zhou et al. 2015). A general overview of the interactions between hydrological, mechanical and ecological processes in frozen grounds is given by Hayashi (2013)."*

*Nevertheless, I would recommend publication after a careful revision, since the paper could provide useful guidelines on how to interpret soil moisture observations in Alpine regions.*

**Specific comments:**
*Abstract*
*Please add more concrete results on which are the main soil moisture patterns.*

The last paragraph of the abstract was modified as follow in order to include more detail about the observed soil moisture patterns.

**Original manuscript:**
*"In this contribution we will present a detailed description of the SOMOMOUNT instrumentation and calibration procedures. Additionally, the data collected during the three first years of the project will be discussed in relation to their altitudinal distribution. Clear differences in soil moisture patterns are visible between sites with permanently and seasonally frozen as well as unfrozen ground conditions and can be related to several factors such as the subsurface composition (organic versus mineral), the elevation and the snow cover characteristics."*

**Revised version:**
*"In this contribution we present a detailed description of the SOMOMOUNT instrumentation and calibration procedures. Additionally, the data collected during the three first years of the project are discussed with regard to their soil type and climate dependency as well as their altitudinal distribution. The observed elevation dependency of soil moisture is found to be non-linear, with an increase of the mean annual values until ~2000m.a.s.l. followed by a decreasing trend towards higher elevations. This altitude threshold marks the change between precipitation/evaporation controlled soil moisture regime and frost affected ones. The former is characterized by high liquid VWC throughout the year and minimum values in summer, whereas the latter typically exhibits long lasting winter minimum liquid VWC values and high variability during the summer.*

*Introduction*
*Line 25-30. The paper seems really focusing on permafrost. Then, later in the results section, the topic permafrost is hardly covered. I suggest a broader motivation. There are other important reasons to monitor mountain soil moisture, from runoff production to climatic impacts to vegetation seasonality.*

The relevant paragraph in the introduction speaking of the importance of soil moisture measurement in mountainous terrains was broadened. Furthermore, the reference to the ground thermal regime was removed from the description of the SOMOMOUNT.

[revised manuscript text omitted]

*Field sites*
*Pages 5-6-7 This part could also become shorter, moving more details to Table 3.*

We thank the reviewer for this comment. The field site description was shortened, and the information relative to the site specific elevation, annual air temperature and precipitation sum have been integrated into table 3.

*Soil moisture temporal evolution. This paragraph is very qualitative, but well written. However, I share here with Reviewer 1 some doubts on the methodology. It is not so straightforward assuming that soil freezes when soil temperatures are negative. Soil could remain unfrozen in small pores also below 0. Also assuming that a TDR device measures low soil moisture when the soil is frozen it is not so obvious. See the comments of the other reviewer on this point.*

We are thankful to the reviewer for this comment. This issue was discussed in detail in the answer to the comments of reviewer 1 (see above).

*Page 11, Line 34. "During the snow melt period only a small VWC increase is seen, which could be attributed to conditions close to saturation throughout the winter." Interesting observation, but it would be nice to quantify better the impact of snow melt on initial VWC in spring. It is a relevant research question in the context of climate change.*

We thank the reviewer for this interesting comment. Unfortunately amongst our field sites only FRE, SCH and STO are equipped with snow sensors. Looking at the data from 2014 at FRE, the same observation can be made as for 2015. In both cases the VWC increase following the total snow melt

was around +2 vol.%. This process is also seen at PLA (a SwissSMEX soil moisture station at 1000m elevation) but with slightly larger VWC increase (+ 5vol.%).

We agree that the quantification of the snow melt contribution to the VWC variation during the spring is an important topic, which would demand a detailed study and is beyond the scope of this paper.

*Page 12, Line 5. Simply stating that VWC is minimum is not correct. The water is in the soil, but likely frozen. It is more correct to specify minimum liquid water content. Therefore, the whole the discussion is quite misleading. You cannot consider in the same way having low liquid (but high total) water content due frozen soil and having a low total water content because the soil is dry. Those are completely different ways to have low liquid water content.*

We thank the reviewer for pointing out this fact – our argumentation was indeed misleading which was not our intention, as we focused throughout the manuscript on the LIQUID water content and not the total water content including ice. In the method section we included a new part that answers the major comments and concerns of the two reviewers concerning the reliability of VWC measurements in frozen conditions (see above). Within this part and throughout the text we now clearly state that whenever the tern VWC is used in the manuscript it always refers to liquid VWC.

*Soil moisture spatial distribution. This paragraph is not very informative. It only informs us that there is (as expected) a large spatial variability, and where falls more rain is more wet. I suggest to skip or strongly reduce.*

We are thankful to the reviewer for this comment. We agree with the reviewer and thus removed the whole section from the manuscript, including Figure 8. Mention of the spatially distributed sensors was added at the end of section 4.2. The main point about the large observed spatial variability is now summarized in the following sentence:

*"These stages are also observed in the data collected by the spatially distributed SMT100 sensors installed at 30cm depth at SCH and STO, with marked differences in absolute VWC as well as variable onset and duration of the three stages even at close vicinity (not shown). "*

*Page 14, Line 28. How thick is the organic layer? What about organic soil in the other sites? Soil type dependency is not discussed for all the sites. Why?*

The characteristics of the organic layer at DRE was clarified in the text as follows:

*"At DRE the soil profile down to 50cm consists of one single organic rich sandy loam layer with a very low bulk density (Table 3), which is underlain by large sized boulders."*

A new paragraph discusses the soil type dependency of the other field sites:

*"At FRE and MLS, the soil type is relatively homogeneous within the uppermost 50cm of the ground, which results in similar VWC temporal evolution with only slight variations in timing and absolute values between the sensors. At MLS the soil type is silty loam and is thus able to retain more water than the sandy loam present at FRE. Finally at SCH and STO, the ground consists of sand or loamy sand, with a significant proportion of large size elements (at 10cm 25% of the soil particles are larger than 10mm at SCH and 45% at STO, see also Fig. 3e-f). Such soil composition is highly heterogeneous even on small distance explaining the high variability between the sensors as well as the comparatively low VWC during unfrozen and snow free periods."*

*Page 15, Line 4. Do you mean Figure 10?*

We are thankful for the comment, which helped to clarify the paragraph. Here we refer to the full dataset of soil moisture, which is shown in Figure 6. The inter-annual variations are then shown in detail in Fig. 10 using the moisture orbit representation. The text was modified to make this distinction more clear.

**Original manuscript:**
*"As seen in Fig. 6, the annual soil moisture dynamics is strongly influenced by the variations in atmospheric conditions such as the extreme temperatures of July 2015."*

**Revised version:**
*"As seen in Fig. 6, the soil moisture temporal evolution is strongly influenced by the variations in atmospheric conditions such as the extreme temperatures of July 2015."*

*Page 16, Line 3 and line 20 "From Fig. 6c, it can be seen that, at the time of the precipitation event, the VWC at 10cm and 30cm depths were unusually low, enabling the water to pass through easily."*
*"As for MLS above, the VWC is low at all depths enabling the precipitations to infiltrate quickly down to the deepest layers"*
*Those statements apparently contradict hydrologic theory. In fact, larger is the water content, larger is the soil hydraulic conductivity and therefore larger is the amount of water that can infiltrate. Please motivate your statements. Do you refer to the speed of the infiltration front or to the total amount of water infiltrated? (i.e. see Green – Ampt theory?). It is not only because if the soil is already wet you do not see a big change in soil moisture, but the infiltration amount is already significant?*

We thank the reviewer for this comment which is of course correct and adapted our interpretation as follows (see also the answer to reviewer 1 above):

*"From Fig. 6c, it can be seen that, at the time of the precipitation event, the VWC at 10cm and 30cm depths were unusually low, thus providing very suitable conditions for the activation of preferential flow (see e.g. Wiekenkamp et al., 2016)."*

*"As for MLS above, the VWC is low at all depths creating suitable conditions for preferential flow. The bypass of the dry uppermost layer and VWC increase first seen at depth is therefore interpreted as being indicative for preferential flow processes at this site"*

*Page 16, Line 18. Do you mean Figure 10?*

No, here we refer to the precipitation event of August 2015, which is shown in detail in Fig.11.No adaptation was made in the text.

*Altitude dependency. This part of the paper is well written and informative, but very qualitative. This altitudinal dependency is already known in literature. Even if it is important to show how the new data published in this paper follow or not follow what is known in literature, a more robust quantitative analysis would be beneficial for the paper.*

In order to assess the altitude dependency of all the selected parameters, we collected all available data from the MeteoSwiss, PERMOS and IMIS (Intercantonal Measurement and Information System, maintained by SLF) networks and performed linear regression of the mean annual (or annual sum) values in 2015 compared to elevation. The results have been included in the new version of Figure 13 and are discussed in the text (see below). Finally, the measured VWC at three additional stations (PAY, PLA and CER) have been added in Figure 13a. Unfortunately the limited number of soil moisture stations available for this study does not allow for more a quantitative analysis of the soil moisture trend with elevation. The new paragraph inserted reads as follows:

*"For each variable, all available data from the monitoring networks of MeteoSwiss, PERMOS and IMIS (Intercantonal Measurement and Information System maintained by the SLF) were collected and a linear regression model was calculated based on the annual mean (resp. sum) of the year 2015. Globally, the same elevation dependency trends are observed using the single stations (dots) or the entire datasets (regression lines)."*

[Figure]

**Fig. 12: Elevation dependency of the winter-, summer- and annual mean liquid VWC at 30cm depth (a), air-, ground temperature and surface offset (ground minus air temperature) at 10cm (b), annual precipitation sum including selected SwissMetNet stations for comparison (cf. Fig. 1) (c), freezing and thawing degree days (calculated from ground temperatures at 10cm) (d) and snow duration (calculated from ground temperature at 10cm using the method described in Staub and Delaloye (2016)) (e) during the year 2015. The VWC values for SIO, PAY and PLA are part of the SwissSMEX network (Mittelbach and Seneviratne, 2012). Due to missing data at 30cm the VWC shown for PAY and PLA was measured at 50cm and 20cm at CER. The dashed green lines illustrate the linear regression based on all available SwissMetNet, PERMOS and IMIS stations in Switzerland (the numbers of station with complete data series in 2015 are indicated) and the shaded areas represent the 99% confidence intervals. The length of each regression line corresponds to the maximum elevation of the available stations.**

*Page 16, Line 28 . "Below 2000 m.a.s.l. the maximum VWC is recorded in winter and the minimum in summer, whereas above this threshold the inverse occurs (maximum VWC in summer and minimum in winter)."*
*This refers only to the liquid water content. The total (frozen + unfrozen) remains high all the winter even at high elevations. Please specify this better.*

We thank the reviewer for pointing out this fact. In the method section we included a new part that answers the major comments and concerns of the two reviewers concerning the reliability of VWC measurements in frozen conditions (see above). Within this part we clearly state that whenever the term VWC is used in the manuscript it always refers to liquid VWC. Furthermore, in cases where confusion might arise, we specified that we are indeed referring to liquid VWC.

*Page 17, Line 5-6 and following. Ok, such processes are somehow clear, but … what are the implications for SWC?*

The implications of the presented elevation trends have been made clearer by modifying the text as follow:

**Original manuscript:**
*"From Fig. 13 the following relationships can be determined: ground- and air temperature, as well as the thawing degree days (absolute sum of positive temperatures per year) all linearly decrease with elevation and conversely the freezing degree days (absolute sum of negative temperatures per year), surface offset and the snow cover duration increase with elevation. There is no clear trend in the precipitation distribution due to the strong microclimatic effects, however on larger spatial scale (continental) precipitation is known to increase with elevation (Smith, 1979)."*

**Revised version:**
*"From Fig. 12 the following relationships can be determined: ground- and air temperature, as well as the thawing degree days (absolute sum of positive temperatures per year) all linearly decrease with elevation, yielding a decreasing trend of evaporation and thus a theoretically increasing trend of liquid VWC. Conversely the freezing degree days (absolute sum of negative temperatures per year), surface offset and the snow cover duration increase with elevation, which results in longer lasting winter liquid VWC minimum and thus lower mean annual liquid VWC with increasing elevation. Finally, a slightly increasing trend in the precipitation distribution is observed with large site specific variations due to the strong microclimatic effects, however on larger spatial scale (continental) precipitation is known to increase with elevation (Smith, 1979). This combination of overall increasing precipitation and decreasing evaporation yields a trend of increasing soil moisture with elevation until the altitude threshold of about 2000m.a.s.l., where it is balanced by the increasingly important ground freezing and soil moisture starts decreasing with elevation."*

*Page 17, Line 18 "This is confirmed by the observed negative ground temperatures as well as the increasing freezing degree days. With increasing elevation, air and ground temperatures decrease yielding increasingly long duration of seasonally frozen ground and thus explaining the decreasing trend of VWC"*
*Could you qualitatively estimate a functional relationship? Which is the temperature threshold? (I have seen that later the paper partially answers to those questions …)*

We thank the reviewer for this interesting question. However, given the all the uncertainties relative to the soil moisture measurements in frost affected soils raised by the reviewers and the limited amount of stations available it is not yet feasible to define a functional relationship. Particularly the lack of soil moisture data at the critical elevation (i.e. between 1600m and 2500m) prevents us from formulating any general relationship. We fully agree that this direction of research should be pursued in further work and we would be very interested for any additional soil moisture data available at these elevations.

*Page 17, lines 30 and followings. Such "frost holes" are geological peculiarities well known in other places of the Alps (I.e. Kaltern, Italy; Lases, Italy). Their effect on SMC is interesting. May be there is more in the geological literature.*

In answer to the comment of reviewer 1 regarding the cooling effect of the air circulation within the talus slope, we clarified the process and highlighted the relative location of our measurement setup. We also included a recent numerical modelling study which was able to reproduce the observed colder ground temperature. To the best of our knowledge none of the studies on this process so far include the effect on the water content.

*Page 18, lines 25-32. Well written, but it could be shortened. Too many unnecessary details.*

We thank the reviewer for this remark and shortened this passage accordingly. It now reads:

**Original manuscript:**
*"From our field sites FRE appears to receive more precipitation than expected, whereas STO receives much less than predicted. Indeed, FRE is situated on top of the easternmost ridge of the Jura mountain chain, which is known to have a comparatively high annual precipitation sum, as is also shown by the neighbouring SwissMetNet stations of Chasseral (1599 m.a.s.l.) and Chaumont (1136 m.a.s.l.), which recorded annual precipitation sums of 1509 mm $y^{-1}$ and 1240 mm $y^{-1}$ respectively for the period 1961-1990 (MeteoSwiss). Precipitation maxima in middle mountain ranges are often found at the highest elevations, whereas the precipitation distribution in high mountain ranges is strongly influenced by the prevailing wind directions and found along the windward slopes, with smaller values on mountain tops and in the lee of the mountains (e.g. Smith 1979). In addition, STO is located in the central alpine region, which is very dry due to the wind shading effects from the surrounding mountain crests. This effect is also seen at the station of Findelen (VSFIN), which is located about 3km from STO and which is also much drier than the stations at comparable elevation (Fig. 13c)."*

**Revised version:**
*"From our field sites FRE appears to receive more precipitation than expected, whereas STO receives much less than predicted. Indeed, FRE is situated on top of the easternmost ridge of the Jura mountain chain, which is known to have a comparatively high annual precipitation sum, as is also shown by the neighbouring SwissMetNet stations of Chasseral (1599 m.a.s.l.) and Chaumont (1136 m.a.s.l.), which recorded annual precipitation sums of 1509 mm $y^{-1}$ and 1240 mm $y^{-1}$ respectively for the period 1961-1990 (MeteoSwiss). STO is located in the central alpine region, which is very dry due to the wind shading effects from the surrounding mountain crest (see e.g. Smith 1979). This effect is also seen at the station of Findelen (VSFIN), which is located about 3km from STO and which is also much drier than the stations at comparable elevation (Fig. 12c)."*

*Conclusions In general, I see here nothing on the implications on permafrost, which seems to be one of the major motivations of the paper. Either downplay this in the introduction, or develop more the permafrost topic in the discussion/conclusions sections.*

We agree with the reviewer on that point and made the according changes in the introduction (see answer above for the specific modifications).

*Page 19, lines 5-12. This part should be moved to the method section. It partially addresses the major objection of Reviewer 1. You say "Although the sensors are not specifically designed for freezing conditions, we found the measurements to be consistent, both regarding inter-sensor comparisons as well as in comparison with related variables such as ground temperature and precipitation". This should be better motivated in the method section.*

In response to the major comment from reviewer 1 and 2, we added a new sub-section after the calibration part, which tackles the issues of VWC measurement in frozen conditions (see above). The selected passage was moved there and the conclusion was modified as follow.

**Original manuscript:**
*"The use of two types of standard soil moisture sensors for application in coarse grained terrain undergoing freeze/thaw cycles at middle and high elevation was shown to be reliable. Although the sensors are not specifically designed for freezing conditions, we found the measurements to be consistent, both regarding inter-sensor comparisons as well as in comparison with related variables such as ground temperature and precipitation."*

**Revised version:**
 *"The use of two types of standard soil moisture sensors for application in coarse grained terrain undergoing freeze/thaw cycles at middle and high elevation was shown to be reliable, both regarding inter-sensor comparisons as well as in comparison with related variables such as ground temperature and precipitation."*

*Page 19, lines 15-16. I suggest to add here more details on the result that it does exist a SWC "peak" at about 2000 m a.s.l. This is an interesting result that should be highlighted.*

An additional sentence was added to the conclusion to highlight the observed 2000m threshold.

*"This shift between the two distinct moisture regimes was found to take place at about 2000m.a.s.l., where the maximum annual VWC values were recorded."*

*Figure 11. I suggest to have the same x and y range in all the subplots.*

We thank the reviewer for this remark. We agree that for the comparison homogeneous scales would be more appropriate. However, given the differential response of each field sites using one single scale makes the orbit too small in some cases (SCH, STO and GFU). Therefore we did not change the scales in figure 11.

---

## Author Comment (AC2) · 12 Dec 2016

We thank the anonymous referee #2 for the very useful and interesting suggestions and corrections. In attachment you can find the one to one answer to the comments of both reviewers.

Please also note the supplement to this comment:
http://www.hydrol-earth-syst-sci-discuss.net/hess-2016-474/hess-2016-474-AC2-supplement.pdf
* * *

---

## Author Response (AR2)

**Suggestions for revision or reasons for rejection (will be published if the paper is accepted for final publication)**

Second review of „ Monitoring soil moisture from middle to high elevation in Switzerland: Set-up and first results from the SOMOMOUNT network" by Pellet et al.

The manuscript has been extensively reworked and most of the earlier reviewer comments have been appropriately addressed. I also very much appreciate the additional chapter on the effects of soil freezing on soil moisture measurements and the restricted accuracy of liquid soil moisture measurements during frozen conditions. This limited measurement accuracy should be also mentioned later in the interpretation and discussion of the results.

However, there are still several issues that need to be resolved before publication can be recommended.

**General comments**

The different terms "liquid VWC", "VWC", "total VWC", "total soil moisture" and "soil moisture" are used at random, which is confusing for the reader. I suggest to consistently using the term "LSM" for liquid soil moisture and "TSM" for total soil moisture (in the following I will use these abbreviations. Also the axis captions of all figures need to be adopted in this way.

We are thankful to the reviewer for this comment. We agree that a consistent use of clear terms such as liquid soil moisture (LSM) and total soil moisture (TSM) would improve the clarity of the text. All the different terms used previously for liquid and total soil moisture in the manuscript have been conscientiously replaced by the terms LSM and TSM following the reviewer's suggestion.

All soil moisture measurements below 0 °C have now been indicated in the figures to indicate frozen soil conditions (i.e. measurements are not representing TSM anymore). However, there are cases where frozen conditions seem to occur above 0 °C. For instance in Fig. 6 (e.g. GFU) one can clearly see that LSM is dropping sharply when temperatures approaches 0 °C, indicating that a substantial part of the soil water within the measurement volume of the sensors has already started to freeze, although soil temperatures is still above 0 °C according to the temperature sensor. This indicates that the temperature measurements are not well representing the measurement volume of the sensor. This is not surprising, since the electromagnetic waves of the soil moistures sensors penetrate a certain part of the soil, whereas the temperature sensor only measures its own temperatures inducing a scale mismatch. In order to circumvent this problem, a higher threshold value for soil temperature should be to be chosen (e.g. 1 °C or higher).

We thank the reviewer for this pertinent comment. Given the accuracy of the temperature sensor (±0.4°C) and the difference of measured volume, we agree with the reviewer that the 0°C threshold is not perfect to indicate frozen/unfrozen soil conditions. Following the reviewer's suggestion a 1°C threshold has been used. All figures have been modified accordingly and an additional mention has been included in the revised manuscript.

**Original manuscript:**
*However, the VWC measurements carried out at temperatures below 0°C are clearly identified in all figures in the manuscript and have to be interpreted with care, especially regarding their absolute values.*

**Revised version:**

*Given the generally lower accuracy of the soil moisture sensors under partially frozen and frozen conditions, the LSM measurements carried out at temperatures below 1°C are clearly identified in all figures hereafter. The 1°C threshold was selected to account for partially frozen conditions as well as the scale mismatch between the temperature and LSM measurements. These data have thus to be interpreted with care, especially regarding their absolute values.*

There are still issues related to the special situation at DRE. Only after reading the recent HESSD paper I was able to understand the process of convective heat transport by air circulation within the talus slope. The authors suggest that is effect also explains lower LSM values. They argue that atmospheric air is transported during winter periods into the ground at the location of the monitoring station due to this process, thus when air temperature is well below 0 °C. This would induce soil freezing below the snow cover and thus explaining the observed drop in LSM. However, the soil temperature stays close to 0 °C during this (Figure 6), which is the typical soil situation below an insulating show cover. In addition the drop in LSM happens mostly in 50 cm, which is counter intuitive since freezing should be more pronounced near the soil surface.

Therefore an alternative explanation for the drop in LSM should be considered: Since the DRE soil has a very high porosity (the bulk density of 0.12 g/cm³ show in Tab. 3 means that the porosity is >90 %!), the drop in LSM could be easily explained by exfiltration of soil water into the bed rock fissures below the soil layer or by lateral water transport downhill. In addition, the drop in LSM in winter is only marginal and cannot explain the substantial lower annual mean LSM compared to MLS or FRE. It is more realistic that LSM at MLS and FRE are higher due to the influence of shallow groundwater that keeps the soil saturated for longer time periods (i.e. LSM stays constant at the maximum value), whereas DRE does not show any sign of groundwater influence (i.e. LSM shows high variability and the LSM is well below the soil porosity).

We are thankful to the reviewer for this important and detailed comment. We agree with the reviewer that the importance of the convective heat transport in our current interpretation of the low LSM at DRE is overestimated. In the dataset presented in this paper the cooling effect only impacts LSM during the winter and over short time period. Thus, it cannot be used to explain the overall low LSM observed, which is rather caused by the specific soil properties.

In the revised version of the manuscript we therefore clarified the respective importance of the soil properties and the convective heat transport process with regard to the mean annual LSM and LSM dynamics. As stated by the reviewer, the overall low mean annual LSM values are due to the soil properties, which lead to water exfiltration and prevents the influence of groundwater. The influence of the air circulation (cooling effect) is restricted to the winter period and explains the prolonged near 0°C temperature observed as well as the corresponding low LSM values. The manuscript was modified as follow in order to clarify this point.

**Original manuscript:**

*To summarise the findings from the SOMOMOUNT network presented above, a simple theoretical model of the evolution of soil moisture and its contributing factors with elevation can be visualised with the grey shading in Figure 14. Comparing the observations qualitatively with this model (circles in Fig. 13), it can be seen that DRE does not fit the model. The recorded mean annual liquid VWC values are much lower than expected.*

*The case of DRE is particular not only for its soil moisture dynamics but also in terms of snow duration (longer than expected) and mean annual ground temperature (lower than expected). Both anomalies are due to site specific characteristics, which are independent from elevation.*

*The low mean annual ground temperature results from…*

In the conclusion:
*Among the six soil moisture stations of SOMOMOUNT, and also in comparison with additional stations from other networks, the station of Dreveneuse is a clear exception to the elevation dependent theoretical model. This middle elevation site undergoes strong winter freezing as well as marked summer evaporation, the latter being due to the vegetation cover. Due to complex air circulation within the underlying talus slope the ground temperatures are unusually low for this elevation. In addition, the soil properties favoured rapid water transport through the ground. The soil properties were found to play an important role in the short term soil moisture variations as well as in the mitigation or intensification of the extreme events.*

**Revised version:**
*To summarise the findings from the SOMOMOUNT network presented above, a simple theoretical model of the distribution of LSM and its contributing factors with elevation can be visualised with the grey shading in Figure 13. Comparing the observations qualitatively with this model (circles in Fig. 13), it can be seen that DRE does not fit the model. The recorded mean annual LSM values are much lower than expected. This is due to the composition of the soil profile. The uppermost layer of the subsurface has a very high porosity (> 90%, see Table 3), which leads to exfiltration of water into the underlying talus slope or lateral water transport. Furthermore, and conversely to FRE and MLS, no evidences of groundwater influence are seen at DRE (Fig. 6). This is consistent with the coarse blocky structure of the talus slope which does not retain the infiltrating water.*

*The case of DRE is also particular in terms of snow duration (longer than expected) and mean annual ground temperature (lower than expected). Both anomalies are due to site specific characteristics, which are independent from elevation. The low mean annual ground temperature results from …*

In the conclusion:
*Among the six soil moisture stations of SOMOMOUNT, and also in comparison with additional stations from other networks, the station of Dreveneuse is a clear exception to the elevation dependent theoretical model. The lower than expected LSM can be attributed to particular soil properties, which favour rapid water transport through the ground. In addition, this middle elevation site undergoes strong winter freezing as well as marked summer evaporation, the latter being due to the vegetation cover. Due to complex air circulation within the underlying talus slope the ground temperatures are unusually low for this elevation. Finally, the soil properties were found to play an important role in the short term LSM variations as well as in the mitigation or intensification of the extreme events.*

In the sensor comparison (Chapter 4.1) only the $R^2$-values are discussed. However, the RMSE is much better indicator for the accuracy of the LSM measurements.

We are thankful to the reviewer for this comment. The reviewer is right of course that the RMSE is the better indicator for accuracy, whereas the R2- values can be used to verify the similarity of the LSM variations. A more in depth discussion of the RMSE is now included in the revised version of the manuscript.

**Original manuscript:**
*At FRE, DRE, MLS and GFU the correlation between the VWC measured by the two sensors was found to be satisfactory (lowest correlation at MLS: $r^2 = 0.749$, Fig. 4).*

And:

*Similar to Fig. 4, the comparison between FDR and TDR sensors at the same depth (Fig. 5) shows a generally good correlation (lowest $r^2 = 0.524$ at STO) with some deviations from the one-to-one relation (black line). At FRE and GFU the comparison between PICO64 and SMT100 soil moisture measurements yield very similar results to the comparison of the two SMT100 sensors, with slightly higher RMSE values. MLS shows a larger dynamic range and mostly higher values for the SMT100 sensor, but a similar temporal variability.*

**Revised version:**
*At FRE, DRE, MLS and GFU the correlation between the LSM measured by the two sensors is found to be satisfactory (lowest correlation at MLS: $r^2 = 0.766$, Fig. 4). The RMSE is more variable with the lowest value at GFU (2.16 vol.%) and the largest at DRE (11.2 vol.%).*

And:

*Similar to Fig. 4, the comparison between SMT100 and PICO64 sensors at the same depth shows a generally good correlation (lowest $r^2 = 0.524$ at STO) with some deviations from the one-to-one relation (Fig. 5). The RMSE is generally larger than for the SMT100 intercomparison (Fig. 4) at all sites, which can be explained by the different measurement volume of the SMT100 and PICO64 sensors (see section 2.1 and Fig. 2). At FRE and GFU the comparison between PICO64 and SMT100 LSM measurements yields very similar results to the comparison of the two SMT100 sensors, with slightly higher RMSE values (4.53 and 2.52 vol.%, respectively). MLS shows a larger dynamic range and mostly higher values for the SMT100 sensor (RMSE = 16.7 vol.%), but a similar temporal variability ($r^2 = 0.606$). A similar pattern is observed, when comparing the PICO64 sensor with the second SMT100 sensor installed at 30cm (Table 5). This dry bias of the PICO64 sensor at MLS is probably due to a bad contact between its rods and the surrounding soil.*

Chapter 3 should be restructured (only one sub-chapter is a bit awkward).

We thank the reviewer for pointing out this fact. The subsection header (3.1 Technical considerations for frozen conditions) was removed in the revised manuscript.

In general, the manuscript should be carefully checked for syntax and tense errors (preferably by a native speaker).

We revised and corrected the whole manuscript again regarding syntax and tense errors. We hope to have adequately addressed all language errors.

**Specific comments**

P1L19: "up to" instead of "until"

We thank the reviewer for pointing out this fact and modified the manuscript accordingly.

**Original manuscript:**
*The observed elevation dependency of soil moisture is found to be non-linear, with an increase of the*

*mean annual values until ~2000m.a.s.l. followed by a decreasing trend towards higher elevations.*

**Revised version:**
*The observed elevation dependency of LSM is found to be non-linear, with an increase of the mean annual values up to ~2000 m.a.s.l. followed by a decreasing trend towards higher elevations.*

P1L21: "VWC" is not defined

Following the general comment that suggested a consistent use of either LSM for liquid soil moisture or TSM for total soil moisture, the term VWC was removed from the revised manuscript.

P3L22: "Qu et al., 2013"
P3L20-21: Actually, the frequency of the SMT100 sensor is not fixed. The SMT100 sensor generates a pulse, which is inverted and then fed back to the input of the line driver resulting in an "oscillation" frequency that mainly depends on the dielectric permittivity of the surrounding medium (between 150 MHz in water and 340 MHz in air), see Bogena et al. (2017) for more details on the SMT100 technology. Bogena et al. (2017) also showed the effects of temperature on SMT100 reading and demonstrated that any temperature dependency of the measured soil moisture are related to temperature related changes in permittivity and thus are not a result of the SMT100 sensor electronics and thus can be easily corrected using temperature information.
P3L26: According to Bogena et al. (2017) the accuracy of the SMT100 for ideal conditions/media is about 1 vol.% (factory calibration) and even better in case of sensor specific calibration.

We thank the reviewer for this detailed comment. The manuscript was modified in order to include this new reference and clarify the working of the SMT100 sensor. Additionally, an external review in the context of the PhD examination pointed out the fact that this sensor uses the frequency domain but is not a reflectometry device. Therefore, the term FDR was removed from the manuscript or replaced by "*frequency domain*".

**Original manuscript:**
*The SMT100 sensors are the newest generation of the so-called SISOMOP sensors, which have been used to monitor soil moisture at Schilthorn (one of the high elevation permafrost sites, see section 2.3) since 2007 (Hilbich et al. 2011), demonstrating the sensors robustness and capability to measure in mountainous areas. Furthermore, laboratory experiments performed by Mittelbach et al. (2012) showed that the SISOMOP sensors have a similar absolute accuracy (±3 vol.%) compared to three other, more widely used, FDR sensors.*

**Revised version:**
*The SMT100 sensors are composed of a ring oscillator which feeds a 10cm long transmission line (Fig. 2). The sensors emit an electromagnetic pulse. Its resulting oscillation frequency is recorded and can then be related to the dielectric permittivity and thus to the LSM of the surrounding medium (see e.g. Bogena et al, 2017; Qu et al., 2013). The SMT100 sensors are the newest generation of the so-called SISOMOP sensors, which have been used to monitor LSM at Schilthorn (one of the high elevation permafrost sites, see section 2.3) since 2007 (Hilbich et al. 2011), demonstrating the sensors robustness and capability to measure in mountainous areas. Additionally, Bogena et al. (2017) showed that the SMT100 sensors have an absolute accuracy of ±1 vol.%  in ideal conditions using the factory calibration.*

P4L4-10: Remove redundancies (e.g. penetration depth)

The manuscript was modified as follow.

**Original manuscript:**
*Finally, the PR2/6 sensor is a 100cm long down-hole water content sensor measuring soil moisture at 6 different depths (10, 20, 30, 40, 60 and 100cm) using the capacitance technique (Fig. 2). Each measurement depth comprises a pair of stainless steel rings, which transmit the 100 MHz electromagnetic signal into the ground, and one detector to record the returned signal. This technique relies on the fact that the emitted wave generates an electromagnetic field, which extends about 100mm into the surrounding soil and depending on its dielectric properties and thus on the VWC is partly reflected (Verhoef et al., 2006). The sensor is lodged in an access polycarbonate tube of 25mm diameter and its measurement volume is ~10cm diameter with an absolute accuracy of ±6 vol.% (Delta-T Device, 2008).*

**Revised version:**
*Finally, the PR2/6 sensor is a 100cm long down-hole water content sensor measuring LSM at 6 different depths using the capacitance technique (Fig. 2). Each measurement depth comprises a pair of stainless steel rings, which transmit the 100 MHz electromagnetic signal into the ground, and one detector, which records the returned signal. The sensor is lodged in an access polycarbonate tube of 25mm diameter and its measurement volume is ~10cm diameter with an absolute accuracy of ±6 vol.% (Delta-T Device, 2008; Verhoef et al., 2006).*

P10L2-4: This statement is not fully correct. Watanabe and Wake (2009) showed that the relationship of liquid water fraction measured with NMR and the permittivity measured with TDR can be approximated with Topp's equation for sand (except −0.1 < T > 0 °C), but not for other soil textures like loam. Thus, for most of your sites this means that the LSM measurements have less accuracy during frozen conditions.

We thank the reviewer for pointing out this fact and modified the manuscript as follow.

**Original manuscript:**
*However, according to Watanabe and Wake (2009), for sand the calibration using Topp's empirical relationship in frozen conditions shows only small deviations from the measured total VWC using NMR except for temperatures between 0 and -1°C.*

**Revised version:**
*For frozen sand however, Watanabe and Wake (2009) showed that TDR devices calibrated using Topp's equation exhibit only small deviations from the measured LSM using NMR except at temperatures between 0 and -1°C.*

P11L3: Check grammar

The manuscript was modified as follow.

**Original manuscript:**
*The TDR-based PICO64 sensors, which have a higher absolute accuracy (Mittelbach et al., 2012) are also, are installed at 30cm depth.*

**Revised version:**
*In addition, the TDR-based PICO64 sensors, having a nominally higher absolute accuracy (Mittelbach*

*et al., 2012) have been installed at 30cm depth.*

P11L8: Here you also should mention the very high RMSE at MLS.

We thank the reviewer for pointing out this fact and modified the manuscript accordingly. See also our response to the general comments.

**Original manuscript:**
*MLS shows a larger dynamic range and mostly higher values for the SMT100 sensor, but a similar temporal variability.*

**Revised version:**
*MLS shows a larger dynamic range and mostly higher values for the SMT100 sensor (RMSE = 16.7 vol.%), but a similar temporal variability ($r^2$ = 0.606).*

P11L21: In fact, the temperatures are typically staying close to 0 °C.

The manuscript was modified as follow.

**Original manuscript:**
*DRE, GFU, SCH and STO show a clear drop of temperatures below the freezing point during the winter, whereas no freezing was recorded at FRE. At MLS negative soil temperatures were only observed at 10cm depth during 10 days in early winter 2016.*

**Revised version:**
*DRE, GFU, SCH and STO exhibit temperatures close to and below the freezing point during the winter, whereas no freezing was recorded at FRE. At MLS negative soil temperatures were only observed at 10cm depth during 10 days in early winter 2016.*

P11L29: A high retention capacity should lead to less variability in temporal soil moisture dynamics.

We thank the reviewer for pointing out this fact and modified the manuscript as follow.

**Original manuscript:**
*At GFU the 10cm sensor is much more variable than the ones at 30 and 50cm and shows much higher values. This is due to the high organic content and high retention capacity of this particular soil layer (Fig.3d and Table 3).*

**Revised version:**
*At GFU the 10cm sensor values are much more variable than the values at 30 and 50cm and exhibit higher values. This is due to the high organic content and low bulk density of this particular soil layer (Fig. 3d and Table 3).*

P11L32: Is the high evaporation rate really only due higher temperature? What about other meteorological parameters, especially low precipitation rates?

We thank the reviewer for pointing out this fact. According to MeteoSwiss (2016) and the measurements available at the different stations, precipitation was also unusually low during this period. We modified the manuscript to include this point.

**Original manuscript:**
*This marked soil moisture decrease is due to the exceptionally high air temperatures recorded in July 2015 (MeteoSwiss, 2016; Scherrer et al., 2016) leading to increased evaporation.*

**Revised version:**
*This marked LSM decrease is due to the exceptionally high air temperatures and low precipitation recorded in July 2015 (MeteoSwiss, 2016; Scherrer et al., 2016) leading to increased evaporation.*

P12L18: This is an indication for preferential flow.

We thank the reviewer for pointing out this fact and added one sentence relative to this point in the manuscript.

**Original manuscript:**
*As expected, the uppermost layer (10cm) reacts stronger than the lower ones to atmospheric forcing, however, the response time is very fast and in some cases almost simultaneous at all depths.*

**Revised version:**
*As expected, the uppermost layer (10cm) reacts stronger to atmospheric forcing than layers below.*

*However, the response time is very fast and in some cases almost simultaneous at all depths, which is an indication for preferential flow.*

P12L29: Why "also"?

The manuscript was modified as follow.

**Original manuscript:**
*During this period liquid VWC increases/decreases slowly but remains decoupled from precipitation events. Punctual lateral inflow and/or snow meltwater infiltration are also possible.*

**Revised version:**
*During this period LSM increases/decreases slowly but remains decoupled from precipitation events. Punctual lateral inflow and/or snow meltwater infiltration are possible.*

P13L13: "event" instead of "daily"

The manuscript was modified accordingly.

**Original manuscript:**
*Using moisture orbits of different time scales (annual, pluri-annual and daily), allows us to analyse the dominant processes playing a role in the temporal evolution of soil moisture.*

**Revised version:**
*Using moisture orbits of different time scales (annual, pluri-annual and event), allows us to analyse the dominant processes playing a role in the temporal evolution of LSM.*

P14L9: Check grammar.
P14L12: Check grammar.

The manuscript was modified as follow.

**Original manuscript:**
*The start of the thawing process at larger depth can be due to preferential water infiltration events or to the influence of warm temperatures from the preceding summer at depth (e.g. Zenklusen Mutter and Phillips, 2012). In the case of the latter, warmer ground temperatures in spring can be found at depth compared to the near surface due to the time-lag of heat propagation into the subsurface. The occurrence of this phenomenon depends on the thermal properties of the subsurface and the strength of the winter freezing. .*

**Revised version:**
*The start of the thawing process at larger depth can be due to preferential water infiltration events or to the influence of high temperatures from the preceding summer (e.g. Zenklusen Mutter and Phillips, 2012). In the latter case, the ground temperatures in spring are higher at depth compared to the near surface due to the time-lag of heat propagation into the subsurface. The occurrence of this phenomenon depends on the thermal properties of the subsurface and the strength of the winter freezing.*

**P14L13: Which depth?**

In this sentence, the potential influence of the warm temperature from the preceding summer on the thawing process is proposed as an explanation for the earlier melt observed at 50cm than at 10cm. Unfortunately no temperature measurements deeper than 50cm are available at that precise location. Thus, the exact depth at which this process occurs is not possible to assess with precision. From the nearby borehole, it is known that the active layer thickness (maximum penetration depth of the 0°C isotherm during the summer) varies between 5m and 10m depending on the year. Furthermore, the process described above has been observed in the borehole at approximatively 1.5m depth in several occasions.

**P14L27: Please indicate the soil type in terms of FAO classification.**

We used the USDA classification throughout the text. For better readability we rephrased the sentence as follows:

[revised manuscript text omitted]

Fig. 6 is overcrowded with time series making it very difficult to read, especially since the colours are also quite similar. Since you are later using only the SMT100 data, I suggest removing all other L. SM data. The 0°C-threshold is not working always (see general comment).

**Revised figure:**

[Figure]

*Fig. 6: SMT100 measured LSM (upper panel) and ground temperatures (lower panel) at each SOMOMOUNT station (a-f). At FRE the uppermost panel displays the PR2/6 measured LSM. The vertical dotted lines at FRE and SCH indicate the period analysed in Fig. 7. The dashed LSM lines represent the soil moisture measurements taken when the ground temperature was below 1°C.*

Fig. 7: Precipitation should be shown for the whole period. Use different axis for snow and LSM (the LSM range is too wide).

**Revised figure:**

[Figure]

*Fig. 7: SMT100 measured LSM, ground temperature and soil resistivity at FRE (a) and SCH (b) from March to August 2015. In addition, daily air temperature, snow depth and precipitation sums are shown as well as the date of the transition between the different stages in the thermal evolution at SCH at 10cm (dashed lines A and B, see text for details). The dashed LSM lines represent the soil moisture measurements taken when the ground temperature was below 1°C.*

Fig. 8: You should use the 10 cm temperature data to indicate freezing conditions.

**Revised figure:**

[Figure]

*Fig. 8: Moisture orbit at each SOMOMOUNT station from the 1st January to the 31st December 2015. The numbered arrows indicate the most important stages at each station as well as the sense of the evolution. The hollow circles represent LSM measurements taken when the temperature was below 1°C at 10cm.*

Fig. 10: Yellow is hardly visible.

**Revised figure:**

[Figure]

*Fig. 10: Moisture orbit at each SOMOMOUNT station for one precipitation event between the 23rd and the 27th August 2015. The LSM values are given as hourly mean and expressed as the change of absolute value compared to the first measurement (23rd August at 23:00). The daily precipitation sums recorded (FRE, DRE and MLS) and extrapolated (GFU, SCH and STO) for the 24th August are indicated.*

**Additional literature**

[revised manuscript text omitted]